# ONLINE LEARNING-GUIDED LEARNING RATE ADAPTATION VIA GRADIENT ALIGNMENT

## ABSTRACT

The performance of an optimizer on large-scale deep learning models depends critically on *fine-tuning* the learning rate, often requiring an extensive grid search over base learning rates, schedules, and other hyperparameters. In this paper, we propose a principled framework called GALA (*Gradient Alignment-based Learning rate Adaptation*), which dynamically adjusts the learning rate by tracking the alignment between consecutive gradients and using a local curvature estimate. Guided by the convergence analysis, we formulate the problem of selecting the learning rate as a one-dimensional online learning problem. When paired with an online learning algorithm such as Follow-the-Regularized-Leader, our method produces a flexible, adaptive learning rate schedule that tends to increase when consecutive gradients are aligned and decrease otherwise. We establish a data-adaptive convergence rate for normalized SGD equipped with GALA in the smooth, nonconvex setting. Empirically, common optimizers such as SGD and Adam, when augmented with GALA, demonstrate robust performance across a wide range of initial learning rates and perform competitively without the need for tuning.

## 1 INTRODUCTION

Stochastic first-order (SFO) methods such as SGD (Robbins & Monro, 1951), AdaGrad (McMahan & Streeter, 2010; Duchi et al., 2011), and Adam (Kingma & Ba, 2015) have been the workhorse for training large-scale models due to their low computational overhead and strong empirical performance. Essentially, the practical performance of SFO methods relies on two components: the choice of base learning rate and how the learning rate evolves during training. The initial selection process is typically done by running a grid search over a range of values, which is referred to as *tuning*. On top of that, the evolution of the learning rate throughout the execution is most commonly done by scaling it externally via a *scheduler*. Depending on the characteristics of the optimizer, the learning rate could also be dynamically updated by some internal mechanism during training.

For instance, SGD is often run with a *constant* learning rate and coupled with a scheduler such as cosine annealing (Loshchilov & Hutter, 2017), linear decay (Defazio et al., 2023) or step decay (Ge et al., 2019) that guides the learning rate following a *predetermined* rule. Similarly, the adaptive methods update the learning rate internally by accumulating the observed gradients based on a prescribed rule that usually tends the learning rate below its initial value. Although optimizers have other parameters such as momentum and weight decay, they are often fixed at the beginning, whereas the learning rate evolves throughout and thus has a larger impact on the final performance.

However, it is unclear how to choose an "empirically viable" combination of base learning rate, optimizer, and scheduler, *a priori*, without tuning over a manually chosen set of parameters. In practice, this issue is compounded by the fact that hyperparameters are often tuned for only a few epochs or on a smaller-scale model, and then directly transferred to large-scale experiments. Yet, it has been documented that early training behaviors can be misleading (Wen et al., 2025) as such configurations will be far from ideal at the true scale. Therefore, this challenge highlights the need for a theoretically principled approach to learning rate adaptation that is robust, flexible, and provable. To put things in perspective, we highlight three key drawbacks of common approaches in this domain.

**Robustness:** Many optimizers, such as SGD and AdaGrad, are highly sensitive to the initial learning rate: an excessively large value can lead to divergence, while a very small one results in stagnation.

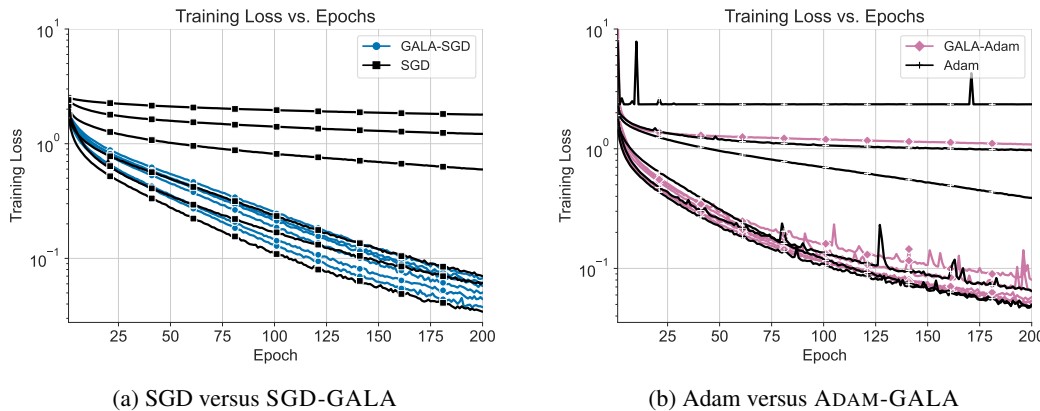

(a) SGD versus SGD-GALA            (b) Adam versus ADAM-GALA

Figure 1: Training loss comparison for standalone SGD and Adam versus GALA applied on their learning rates (SGD-GALA and ADAM-GALA, respectively). The curves are obtained by running the algorithms with initial learning rates $[1, 10^{-1}, 10^{-2}, 10^{-3}, 10^{-4}, 10^{-5}]$.

Ideally, we seek stable performance across a wide range of initial learning rates, allowing for robust training even with suboptimal initializations.

**Adaptivity:** In most cases, the value of the learning rate, either through internal dynamics or an external scheduler, tends to decay over time, limiting flexibility. For instance, the standard AdaGrad algorithm (McMahan & Streeter, 2010; Duchi et al., 2011) reduces the learning rate below its initial value, and most commonly used schedulers induce decaying behavior on top of the learning rate. A more desirable alternative would be a principled and adaptive scheduling mechanism capable of both increasing and decreasing the learning rate as needed.

**Principled:** Most theoretical frameworks in this area are grounded in convex optimization. For instance, AdaGrad and its variants are supported by data-dependent regret bounds derived from online convex optimization. However, the decaying nature of their learning rate is not necessarily empirically optimal for non-convex landscapes. Developing a theoretically grounded approach tailored to non-convex problems is crucial for establishing provable performance guarantees.

While several existing works partially address these three limitations, none meet all the desired criteria simultaneously, as discussed in detail in Section 1.1.

**Our contributions:** Our goal is to unify all three ingredients in a single principled framework. To this end, we propose **G**radient **A**lignment-based **L**earning rate **A**daptation (GALA), an online-learning guided framework for adjusting the learning rate on-the-fly by carefully monitoring the evolution of the optimization path. Our main goal with designing this framework is to **make algorithms robust to parameter-tuning** while **maintaining competitive performance**. In particular:

1. Motivated by the convergence analysis, we construct a one-dimensional online loss function for selecting the learning rate, using the *alignment between consecutive stochastic gradients* and a *local estimate of the gradient Lipschitz constant*. The learning rate is then updated by performing a step for the one-dimensional online problem via any suitable algorithm.

2. Our approach enables dynamic learning rate adaptation: it tends to *increase* when the gradients are aligned and *decrease* when misalignment is detected. We carefully moderate the alignment signal using a regularization term based on the local gradient Lipschitz estimate, promoting stability.

3. Theoretically, we provide a regret-based analysis and establish convergence guarantees for a variant of our algorithm for nonconvex objective functions with stochastic gradients.

4. Empirically, our method demonstrates strong, stable performance across a wide range of hyperparameter settings. We propose a heuristic implementation for other SFO methods.

In fact, Figure 1 provides a glimpse at the performance of our framework when applied to the learning rates of SGD and Adam. We show that GALA helps mitigate sensitivity to the initialization of the

learning rate while maintaining a competitive performance with respect to the best-performing runs of the standalone SGD and Adam.

## 1.1 RELATED WORK

**Classical stochastic first-order methods.** Dating back to the seminal work Robbins & Monro (1951), the theoretical behavior of SGD and its many variants have been extensively studied. Considering general smooth functions, it is well-known that the learning rate must decrease at a rate of $\eta_t = O(1/\sqrt{t})$ where $t$ is the iteration counter and also satisfy $\eta_t \leq O(1/L)$. Ghadimi & Lan (2013) established that SGD with a properly chosen learning rate achieves a complexity of $\mathcal{O}(\epsilon^{-2} + \sigma^2 \epsilon^{-4})$, interpolating between deterministic and stochastic rates and matching the lower bounds (Arjevani et al., 2023). However, the choice of the learning rate depends on the problem parameters, i.e., $L, \sigma$, which are typically unknown and prohibitively difficult to estimate in practice. Similar requirements are in place for the learning rate when the objective function is $\rho$-weakly convex (Davis et al., 2020).

**Adaptive and parameter-free optimization methods.** AdaGrad was introduced in two concurrent works McMahan & Streeter (2010); Duchi et al. (2011) for minimizing a sequence of *online* convex losses. The main idea is to compute a time-varying learning rate by accumulating squared norms of stochastic gradients. This fundamental idea paved the way for many algorithms such as Adam (Kingma & Ba, 2015), RMSProp Tieleman & Hinton (2012), Adadelta Zeiler (2012), and their variants, which demonstrate strong empirical performance. Beyond the online optimization setup, they have been shown to automatically adapt to problem-dependent parameters such as smoothness, noise variance, and bounds on gradients. Their convergence properties have been well-studied for the convex setting (Levy, 2017; Levy et al., 2018; Kavis et al., 2019; Joulani et al., 2020; Antonakopoulos et al., 2022; Liu et al., 2023a; Rodomanov et al., 2024) and non-convex setting (Li & Orabona, 2019; Ward et al., 2020; Li & Orabona, 2020; Kavis et al., 2022; Gadat & Gavra, 2022; Faw et al., 2022; Attia & Koren, 2023; Liu et al., 2023b).

A downside of the first-generation adaptive methods is the sensitivity to initial learning rate due to the need to know an estimate to the initial distance, $\|\mathbf{x}_0 - \mathbf{x}^*\|$. To remedy this, *parameter-free* optimization (Carmon & Hinder, 2022; Ivgi et al., 2023; Khaled et al., 2023; Kreisler et al., 2024; Attia & Koren, 2024) has gained popularity with a focus on augmenting robustness. Essentially, they combine AdaGrad-type learning rate with a certificate that estimates the initial distance to the solution. Although this helps increase from the initial value, the scaling factor is practically bounded, restricting flexibility. On a related front, a different line of work Malitsky & Mishchenko (2020; 2024); Li & Lan (2024) studied parameter-free gradient methods with local curvature estimation for convex, deterministic problems. They are separated from AdaGrad-type methods with a non-monotone learning rate that estimates the time evolution of local smoothness. A downside to these methods is empirical stability; if the increasing behavior is not tamed properly, optimization performance could be unstable, especially for nonconvex problems. Therefore, it is of utmost importance to strike the right balance between flexibility and stability.

**Hypergradient descent.** Originally proposed as a heuristic for stochastic optimization in Almeida et al. (1999), hypergradient descent updates the learning rate by computing the gradient with respect to the learning rate. It was later rediscovered and adapted to modern deep learning Rubio (2017); Baydin et al. (2018), with several subsequent works refining this approach Chandra et al. (2022); Ozkara et al. (2024). Recently, Gao et al. (2024); Chu et al. (2025) provided convergence guarantees from an online learning perspective, though their analysis is limited to deterministic convex settings.

**Online learning-guided methods.** Drawing insights from parameter-free online learning Orabona & Pál (2016), Orabona & Tommasi (2017) reformulate SGD as a coin-betting game and apply a betting algorithm to eliminate the need for a manually tuned learning rate. They also provide convergence guarantees for convex and quasi-convex objectives. Cutkosky et al. (2023a) proposed a general technique for adaptively scaling any base optimization algorithm and learning rate schedule, which is grounded in a black-box reduction framework from parameter-free online learning Cutkosky & Orabona (2018). The work most relevant to ours is that of Zhuang et al. (2019), who consider non-convex stochastic optimization and introduce a surrogate loss technique for selecting the learning rate. However, their method requires knowledge of problem-dependent parameters (e.g., gradient's Lipschitz constant), which limits its flexibility.

## 2 PRELIMINARIES

We consider the stochastic optimization problem

$$\min_{\mathbf{x} \in \mathbb{R}^d} F(\mathbf{x}) = \mathbb{E}_{\xi \sim \mathcal{D}}[f(\mathbf{x}; \xi)],$$

where $f(\cdot; \xi)$ is a random function indexed by a random variable $\xi$ drawn from distribution $\mathcal{D}$. The objective function $F : \mathbb{R}^d \to \mathbb{R}$ is assumed to be differentiable, possibly nonconvex and bounded from below, i.e., $F(\mathbf{x}) > -\infty$. Moreover, we make the following standard assumptions:

**Assumption 1.** *The gradient of $F$ is $L$-Lipschitz continuous, i.e., $\|\nabla F(\mathbf{x}) - \nabla F(\mathbf{y})\| \leq L\|\mathbf{x} - \mathbf{y}\|$ for any $\mathbf{x}$ and $\mathbf{y}$.*

**Assumption 2.** *The stochastic gradient has bounded variance of $\sigma^2$, i.e., $\mathbb{E}[\|\nabla F(\mathbf{x}) - \nabla f(\mathbf{x}; \xi)\|^2] \leq \sigma^2$ for any $\mathbf{x} \in \mathbb{R}^d$.*

### 2.1 BACKGROUND: ONLINE LEARNING

Let us briefly introduce the online learning framework and establish the groundwork necessary within the context of our approach. In the online learning framework, a learner makes decisions iteratively over rounds. At each round $t = 1, \cdots, T$:

1. The learner makes a decision $\mathbf{x}_t \in \mathcal{X}$ from a bounded set of actions;

2. The environment/adversary reveals the loss function $\ell_t(\cdot)$;

3. The learner suffers the loss $\ell_t(\mathbf{x}_t)$.

The learner chooses its action $\mathbf{x}_t$ in round $t$ *prior to* observing the loss $\ell_t(\cdot)$. The performance of the learner is measured by *regret*, which is defined as the difference between the cumulative loss of the learner compared against a fixed action $\mathbf{x}$:

$$\text{Reg}_T(\mathbf{x}) = \sum_{t=1}^{T} (\ell_t(\mathbf{x}_t) - \ell_t(\mathbf{x})). \tag{1}$$

The goal is to achieve *sublinear regret*, i.e., $\text{Reg}_T(\mathbf{x}) = o(T)$, such that the time average of regret goes to zero as $T \to \infty$, meaning the learner performs as well as the fixed strategy in the limit.

## 3 ONLINE LEARNING RATE SELECTION

We begin by introducing a simplified template that outlines our design. Our primary goal is to provide insight into the idea of gradient alignment, explain our adaptive strategy, and establish the foundation for the online learning formulation of the learning rate. Consider the SGD update rule

$$\mathbf{x}_{t+1} = \mathbf{x}_t - \eta_t \mathbf{g}_t(\mathbf{x}_t), \quad \mathbf{g}_t(\mathbf{x}_t) = \nabla f(\mathbf{x}_t; \xi_t), \tag{2}$$

where $\xi_t \sim \mathcal{D}$ is a random sample drawn from the distribution $\mathcal{D}$ at iteration $t$. Our goal is to choose a sequence of learning rates guided by the progress of the algorithm, as measured by the function value difference $F(\mathbf{x}_{t+1}) - F(\mathbf{x}_t)$. At this point, we deviate from the classical analysis; inspired by Cutkosky et al. (2023b), we apply the fundamental theorem of calculus to get

$$F(\mathbf{x}_{t+1}) - F(\mathbf{x}_t) = \langle \boldsymbol{\nabla}_t, \mathbf{x}_{t+1} - \mathbf{x}_t \rangle = -\eta_t \langle \boldsymbol{\nabla}_t, \mathbf{g}_t(\mathbf{x}_t) \rangle, \tag{3}$$

where $\boldsymbol{\nabla}_t = \int_0^1 \nabla F(\mathbf{x}_t + \lambda(\mathbf{x}_{t+1} - \mathbf{x}_t)) \, d\lambda$ denotes the average gradient along the line segment between $\mathbf{x}_t$ and $\mathbf{x}_{t+1}$. Note that the right-hand side of (3) concerns the *alignment* between the gradients $\boldsymbol{\nabla}_t$ and $\mathbf{g}_t(\mathbf{x}_t)$ and serves as a useful signal for adjusting the learning rate. When the alignment term is positive, it indicates that the gradients point in similar directions and increasing the learning rate may lead to greater progress. Conversely, a negative alignment implies opposing directions, in which case a smaller learning rate may be more appropriate.

However, computing $\boldsymbol{\nabla}_t$ is generally intractable, as it involves the true gradient and an integral. A key observation in Zhang et al. (2020); Cutkosky et al. (2023b) is that an unbiased estimate of $\boldsymbol{\nabla}_t$ can be constructed by evaluating the gradient at a random point along the line segment. Specifically, let $\lambda_t$ be a random variable uniformly distributed over $[0, 1]$, and let $\xi'_t$ be an independent sample

---

**Algorithm 1:** SGD-GALA

---

**Input:** Initial point $\mathbf{x}_0$, initial learning rate $\eta_0$, maximum learning rate $\eta^{\max}$, $\delta > 0$

1 **for** $t = 0$ **to** $T$ **do**
2      Sample $\xi_t \sim \mathcal{D}$ and compute $\mathbf{g}_t(\mathbf{x}_t) = \nabla f(\mathbf{x}_t; \xi_t)$
3      $\mathbf{x}_{t+1} = \mathbf{x}_t - \eta_t \mathbf{g}_t(\mathbf{x}_t)$
4      Sample $\xi_t' \sim \mathcal{D}$ and compute $\mathbf{g}_t'(\mathbf{x}_t) = \nabla f(\mathbf{x}_t; \xi_t')$
5      Sample $\mathbf{s}_t \sim \text{Uniform}[0,1]$, compute $\mathbf{w}_t = \mathbf{x}_t + s_t(\mathbf{x}_{t+1} - \mathbf{x}_t)$ and $\mathbf{g}_t'(\mathbf{w}_t) = \nabla f(\mathbf{w}_t; \xi_t')$
6      Compute $L_t = \frac{\|\mathbf{g}_t'(\mathbf{w}_t) - \mathbf{g}_t'(\mathbf{x}_t)\|}{\|\mathbf{w}_t - \mathbf{x}_t\|}$
7      $\eta_{t+1} = \text{clip}_{[0, \eta^{\max}]} \left( \frac{\sum_{s=0}^{t} \langle \mathbf{g}_s'(\mathbf{w}_s), \mathbf{g}_s(\mathbf{x}_s) \rangle}{\delta + \sum_{s=0}^{t} L_s \|\mathbf{g}_s(\mathbf{x}_s)\|^2} \right)$
8 **end**

---

from the distribution $\mathcal{D}$. Then for $\mathbf{w}_t = \mathbf{x}_t + \lambda_t(\mathbf{x}_{t+1} - \mathbf{x}_t)$ and $\mathbf{g}_t'(\mathbf{w}_t) = \nabla f(\mathbf{w}_t; \xi_t')$, we have $\boldsymbol{\nabla}_t = \mathbb{E}_{\lambda_t}[\nabla F(\mathbf{w}_t)] = \mathbb{E}_{\lambda_t, \xi_t'}[\mathbf{g}_t'(\mathbf{w}_t)]$, which implies

$$F(\mathbf{x}_{t+1}) - F(\mathbf{x}_t) = -\eta_t \, \mathbb{E}_{\lambda_t, \xi_t'}[\langle \mathbf{g}_t'(\mathbf{w}_t), \mathbf{g}_t(\mathbf{x}_t) \rangle]. \tag{4}$$

To maximize the decrease in the function value, Eq. (4) suggests that a natural objective is to minimize $-\eta_t \, \mathbb{E}_{\lambda_t, \xi_t'}[\langle \mathbf{g}_t'(\mathbf{w}_t), \mathbf{g}_t(\mathbf{x}_t) \rangle]$. However, this approach comes with two issues. Let us begin with the first point, which is related to the convergence metric. This approach only leads to an upper bound on $\frac{1}{T} \sum_{t=0}^{T-1} \mathbb{E}[\langle \mathbf{g}_t'(\mathbf{w}_t), \mathbf{g}_t(\mathbf{x}_t) \rangle]$, which does not directly provide a meaningful bound on gradient norm of $F$, which is the standard metric we would like to obtain. Our idea is to decompose the inner product as $\langle \mathbf{g}_t'(\mathbf{w}_t), \mathbf{g}_t(\mathbf{x}_t) \rangle = \langle \mathbf{g}_t'(\mathbf{x}_t), \mathbf{g}_t(\mathbf{x}_t) \rangle + \langle \mathbf{g}_t'(\mathbf{w}_t) - \mathbf{g}_t'(\mathbf{x}_t), \mathbf{g}_t(\mathbf{x}_t) \rangle$, where $\mathbf{g}_t'(\mathbf{x}_t) = \nabla f(\mathbf{x}_t; \xi_t')$. Note that the first term leads to $\mathbb{E}[\langle \mathbf{g}_t'(\mathbf{x}_t), \mathbf{g}_t(\mathbf{x}_t) \rangle] = \mathbb{E}[\|\nabla F(\mathbf{x}_t)\|^2]$ and the second term can be controlled using the Lipschitz constant of the gradient.

The second issue is that the minimization of the right-hand side of Eq. (4) with respect to the learning rate $\eta_t$ is an *implicit* problem. The objective depends on the interpolated point $\mathbf{w}_t$, which could be determined only *after* the learning rate $\eta_t$ is chosen. The solution is to cast the learning rate selection as an *online learning problem*, and derive a sequence of online loss functions that will govern the selection process. We combine and formalize both ideas in the following lemma.

**Lemma 1.** *Define the local gradient Lipschitz estimate* $L_t = \frac{\|\mathbf{g}_t'(\mathbf{w}_t) - \mathbf{g}_t'(\mathbf{x}_t)\|}{\|\mathbf{w}_t - \mathbf{x}_t\|}$ *and the surrogate loss function*

$$\ell_t(\eta) \triangleq -\eta \langle \mathbf{g}_t'(\mathbf{w}_t), \mathbf{g}_t(\mathbf{x}_t) \rangle + 0.5 L_t \|\mathbf{g}_t(\mathbf{x}_t)\|^2 \eta^2. \tag{5}$$

*Suppose that Assumption 2 holds and* $L_t \leq L^{\max}$ *for any* $t \geq 0$ *with probability one. Then we have*

$$\sum_{t=0}^{T-1} \mathbb{E}[(\eta - \eta^2 L^{\max}) \|\nabla F(\mathbf{x}_t)\|^2] \leq \mathbb{E}[F(\mathbf{x}_0) - F(\mathbf{x}_T) + L^{\max} T \eta^2 \sigma^2 + \sum_{t=0}^{T-1} (\ell_t(\eta_t) - \ell_t(\eta))]. \tag{6}$$

As shown in (5), our surrogate loss function $\ell_t(\eta)$ consists of two terms. The first term measures the *alignment* between two consecutive (stochastic) gradients $\mathbf{g}_t'(\mathbf{w}_t)$ and $\mathbf{g}_t(\mathbf{x}_t)$, and the second term is a quadratic regularization term that depends on our local estimate of the Lipschitz constant $L_t$. The online nature of the problem is due to the fact that both $\mathbf{g}'(\mathbf{w}_t)$ and $L_t$ can only be computed after the learning rate $\eta_t$ is chosen. Moreover, $\eta$ in (6) is the comparator of our online learning problem and it can be chosen *arbitrarily* in our analysis. If we achieve a low regret for the online learning problem (as we show in Section 4), then a proper choice of $\eta$ will lead to a complexity of $\mathcal{O}(\epsilon^{-2} + \sigma^2 \epsilon^{-4})$.

*Remark* 1. Our approach is inspired by both Cutkosky et al. (2023b) and Zhuang et al. (2019). Compared to Cutkosky et al. (2023b), the key difference is that their method uses online learning to guide the choice of the update direction, whereas we focus on selecting the learning rate. In contrast to Zhuang et al. (2019), our method differs in two major ways: (i) we estimate the local gradient Lipschitz constant on the fly, instead of relying on a global Lipschitz estimate; and (ii) for the first term, we use the alignment between two stochastic gradients evaluated at different points $\mathbf{w}_t$ and $\mathbf{x}_t$, while Zhuang et al. (2019) use gradients at the same point.

The next step is using an online learning algorithm that will operate on the loss sequence $\ell_t$ to update our learning rate $\eta_t$. Since the loss functions are quadratic in their input, we have several options to

choose from. As an example, the Follow-the-Regularized-Leader (FTRL) algorithm is given by

$$\eta_{t+1} = \arg\min_{\eta \in [0, \eta^{\max}]} \left\{ \sum_{s=0}^{t} \ell_s(\eta) + \frac{\delta}{2}\eta^2 \right\},$$

where $\eta^{\max}$ is the maximal learning rate and $\delta \geq 0$ is a user-defined constant to ensure stability. Using the loss in (5) and the FTRL update, we obtain a closed-form expression for $\eta_{t+1}$:

$$\eta_{t+1} = \text{clip}_{[0, \eta^{\max}]} \left( \frac{\sum_{s=0}^{t} \langle \mathbf{g}'_s(\mathbf{w}_s), \mathbf{g}_s(\mathbf{x}_s) \rangle}{\delta + \sum_{s=0}^{t} L_s \|\mathbf{g}_s(\mathbf{x}_s)\|^2} \right), \tag{7}$$

where $\text{clip}_{[0, \eta^{\max}]}(\cdot)$ denotes the operation that clips a real-valued input to the interval $[0, \eta^{\max}]$.

*Remark* 2. The learning rate incorporates directional information along the optimization path through the alignment term: when the gradients are aligned, the learning rate is encouraged to increase; when they are misaligned, it decreases. Additionally, the quadratic regularization term moderates the learning rate update based on the magnitude of observed gradients. Note that this adaptive behavior is an inherent feature of our online learning-guided learning rate and holds by default for various choices of online learners, such as OGD (Zinkevich, 2003; Orabona, 2019).

*Remark* 3. For numerical stability, we pick FTRL as our choice of online learner to update the learning rate of the algorithm but FTL is also applicable since the surrogate losses are quadratic. In either case, the resulting update for the learning rate is independent of the initialization; only the very first step is taken using the base learning rate.

## 4 CONVERGENCE ANALYSIS

In this section, we analyze a variant of our proposed method in Algorithm 1 and establish its convergence rate for stochastic nonconvex optimization. Instead of using the standard SGD update rule in (2), we adopt the normalized SGD with momentum (Cutkosky & Mehta, 2020). As we will show, the main theoretical advantage of using a normalized update is that it simplifies the surrogate loss function, making the regret bound easier to establish. Specifically, we consider the update rule

$$\mathbf{x}_{t+1} = \mathbf{x}_t - \eta_t \mathbf{m}_t / \|\mathbf{m}_t\|, \qquad \mathbf{m}_t = (1 - \alpha)\mathbf{m}_{t-1} + \alpha \nabla f(\mathbf{x}_t; \xi_t), \tag{8}$$

where $\alpha \in (0, 1]$ is the momentum parameter. The normalization step ensures that the learning rate $\eta_t$ directly controls the distance between $\mathbf{x}_{t+1}$ and $\mathbf{x}_t$, thus promoting stability. However, normalization can also amplify the noise in the stochastic gradient. To mitigate this, we apply an exponential moving average in (8), which reduces variance and is governed by the momentum parameter $\alpha$. Due to the different update rule in (8), the surrogate loss function must be modified accordingly. Specifically, we define a new surrogate loss function as

$$\ell_t^{\text{N}}(\eta) = -\eta \left\langle \mathbf{g}'_t(\mathbf{w}_t), \frac{\mathbf{m}_t}{\|\mathbf{m}_t\|} \right\rangle + \left( \frac{1}{2}\max\{L_t, M\} + \frac{4(1-\alpha)\tilde{L}_t}{3\alpha} \right)\eta^2, \tag{9}$$

where $M > 0$ is a user-defined constant and $\tilde{L}_t = \frac{\|\mathbf{g}'_t(\mathbf{x}_{t+1}) - \mathbf{g}'_t(\mathbf{x}_t)\|}{\|\mathbf{x}_{t+1} - \mathbf{x}_t\|}$ is a second local gradient Lipschitz estimate. There are three main differences compared with (5). First, the linear term in (9) measures the alignment between the gradient $\mathbf{g}'_t(\mathbf{w}_t)$ and the normalized update direction $\frac{\mathbf{m}_t}{\|\mathbf{m}_t\|}$, rather than with the stochastic gradient $\mathbf{g}_t(\mathbf{x}_t)$. In addition, the quadratic term is independent of the norm of the stochastic gradient $\|\mathbf{g}_t(\mathbf{x}_t)\|$ due to the normalization step. Moreover, it includes an additional regularization term that depends on the momentum parameter $\alpha$ and the local gradient Lipschitz estimate $\tilde{L}_t$, which arises from the analysis of momentum. In the following theorem, we establish the convergence rate of the update rule in (8) in terms of the regret with respect to the new surrogate loss in (9). The proof can be found in Appendix C.

*Remark* 4. The quantity $M$ is introduced to ensure that the surrogate loss function in (9) is $M$-strongly convex, to facilitate the regret analysis later in Lemma 2. This is also a design choice to ensure numerical stability in a similar spirit to the $\delta$ parameter in Adam step size.

**Theorem 1.** *Let $\{\mathbf{x}_t\}_{t=0}^{T-1}$ be generated by the update rule in (8) and $\eta_t \leq \eta^{\max}$ for all $t$. Recall that $L_t = \frac{\|\mathbf{g}'_t(\mathbf{w}_t) - \mathbf{g}'_t(\mathbf{x}_t)\|}{\|\mathbf{w}_t - \mathbf{x}_t\|}$, $\tilde{L}_t = \frac{\|\mathbf{g}'_t(\mathbf{x}_{t+1}) - \mathbf{g}'_t(\mathbf{x}_t)\|}{\|\mathbf{x}_{t+1} - \mathbf{x}_t\|}$, and the surrogate loss $\ell_t^{\text{N}}(\eta)$ defined in (9). Moreover, define the regret $\text{Reg}_T^{\text{N}} \triangleq \max_{\eta \in [0, \eta^{\max}]} \sum_{t=0}^{T-1} (\ell_t^{\text{N}}(\eta_t) - \ell_t^{\text{N}}(\eta))$, the*

*initial function value gap* $\Delta_F \triangleq F(\mathbf{x}_0) - F(\mathbf{x}^*)$, *the average Lipschitz estimate* $L_T^{\mathrm{avg}} = \max\left\{\mathbb{E}\left[\frac{1}{T}\sum_{t=0}^{T-1}\max\{L_t, M\}\right], \mathbb{E}\left[\frac{1}{T}\sum_{t=0}^{T-1}\tilde{L}_t\right]\right\}$. *Then for* $\alpha = \min\{\frac{1}{\sigma\sqrt{T}}, 1\}$, *it holds that*

$$\frac{1}{T}\sum_{t=0}^{T-1}\mathbb{E}[\|\nabla F(\mathbf{x}_t)\|] = \mathcal{O}\Big(\frac{\sigma^{1/2}(L_T^{\mathrm{avg}}(\Delta_F + \mathbb{E}[\mathrm{Reg}_T^{\mathrm{N}}]))^{1/2}}{T^{1/4}} + \frac{\sigma^2}{\sqrt{T}} + \frac{\sqrt{L_T^{\mathrm{avg}}(\Delta_F + \mathbb{E}[\mathrm{Reg}_T^{\mathrm{N}}])}}{\sqrt{T}}$$
$$+ \frac{\Delta_F + \mathbb{E}[\mathrm{Reg}_T^{\mathrm{N}}]}{\eta^{\max}T}\Big).$$

*Remark* 5. In Theorem 1, the choice of $\alpha$ is made to obtain the best dependence on both $T$ and $\sigma$. We note that assuming knowledge of $T$ is standard in the literature and typically not restrictive in practice. If $\sigma$ is unknown, one could choose $\alpha = 1/\sqrt{T}$ and still obtain the same convergence rate, albeit with a worse dependence on $\sigma$.

The rate in Theorem 1 depends on the regret of the associated online learning problem. Hence, we propose to use an online algorithm to adaptively update the learning rate $\eta_t$ in (8). Note that the loss function in (9) is strongly convex and thus logarithmic regret is possible. For best theoretical guarantees, we use optimistic FTRL (Rakhlin & Sridharan, 2013; Steinhardt & Liang, 2014; Mohri & Yang, 2016):

$$\eta_{t+1} = \underset{\eta \in [0, \eta^{\max}]}{\arg\min}\left\{\sum_{s=0}^{t}\ell_t(\eta) + \frac{\delta}{2}\eta^2 + h_{t+1}(\eta)\right\}, \tag{10}$$

where $h_{t+1}(\cdot)$ is a hint function that aims to approximate the next loss function $\ell_{t+1}$. Specifically, note that $\mathbf{m}_{t+1}$ is already known at the time when we select $\eta_{t+1}$. Hence, we set $h_{t+1}(\eta) = -\eta\langle\mathbf{g}_{t+1}(\mathbf{x}_{t+1}), \frac{\mathbf{m}_{t+1}}{\|\mathbf{m}_{t+1}\|}\rangle$, which yields the following closed-form update rule:

$$\eta_{t+1} = \mathrm{clip}_{[0, \eta^{\max}]}\Big(\frac{\sum_{s=0}^{t}\langle\mathbf{g}_s'(\mathbf{w}_s), \mathbf{m}_s/\|\mathbf{m}_s\|\rangle + \langle\mathbf{g}_{t+1}(\mathbf{x}_{t+1}), \mathbf{m}_{t+1}/\|\mathbf{m}_{t+1}\|\rangle}{\delta + \sum_{s=0}^{t}(\max\{L_s, M\} + \frac{8(1-\alpha)}{3\alpha}\tilde{L}_s)}\Big).$$

We bound the regret of the above update rule in the following lemma (see Appendix D for the proof).

**Lemma 2.** *Let* $\eta^{\max} = \sqrt{\alpha}\bar{\eta}$ *for some given* $\bar{\eta}$. *Suppose that* $\max\{L_t, \tilde{L}_t\} \leq L^{\max}$ *hold for any* $t$ *with probability one. Then we have*

$$\mathbb{E}[\mathrm{Reg}_T^{\mathrm{N}}] = \mathcal{O}\Big(\bar{\eta}^2\max\{L^{\max}, M\}\log\Big(1 + \frac{\max\{L^{\max}, M\}}{\alpha\delta}T\Big) + \frac{\sigma^2}{M}\log T\Big).$$

*Remark* 6. (*Dependence on* $\eta_0$). Note that the convergence rate in Theorem 1 and the regret bound in Lemma 2 do not explicitly depend on the initial learning rate $\eta_0$; instead, $\eta_0$ influences the bound indirectly via the initial gradient Lipschitz estimates $L_0, \tilde{L}_0$. As a result, our method is robust to a wide range of initializations, as demonstrated in the experiments in Section 5. In contrast, the convergence rate of SGD has the form $\frac{\Delta_F + \eta_0^2\sigma^2}{\eta_0\sqrt{T}}$, so when $\eta_0$ is too small/large *multiplicatively* slows down the rate.

Combining Theorem 1 and Lemma 2, up to logarithmic factors, we have established that our method achieves a convergence rate of $\mathcal{O}(\frac{\sigma^{1/2}}{T^{1/4}} + \frac{1}{\sqrt{T}})$, which matches the rate in Cutkosky & Mehta (2020) with a constant learning rate. Moreover, the convergence rate in Theorem 1 is in terms of the average and maximum Lipschitz estimates, which can be much smaller than the global constant $L$ in Cutkosky & Mehta (2020). Notably, we achieve this by adaptively selecting the learning rate instead of using a predefined constant. Finally, we remark that our convergence results are comparable to those obtained for AdaGrad (Faw et al., 2022; Attia & Koren, 2023; Liu et al., 2023b), with the key distinction that our learning rate can *both increase and decrease*, while the AdaGrad rate is monotonically decreasing.

*Remark* 7. (*Role of* $\eta_{max}$). The parameter $\eta_{\max}$ is introduced primarily for theoretical purposes, allowing us to apply FTRL and its standard regret guarantee. In practice, we remove the clipping step in Eq. (7), and hence $\eta_{\max}$ is *not required* as a hyperparameter in our experiments.

## 5 NUMERICAL EXPERIMENTS

We present our results of applying GALA on training a residual network (He et al., 2016) on the CIFAR-10, CIFAR-100 (Krizhevsky, 2009) and Flower102 (Nilsback & Zisserman, 2008) as well as

a Vision Transformer (Dosovitskiy et al., 2021) on Tiny-ImageNet (Stanford CS231n, 2015). We begin with the practical considerations we adopted for efficiency and performance of the algorithms.

## 5.1 IMPLEMENTATION DETAILS

**SGD with GALA.** When applying GALA to augment the standard SGD update rule, we introduce the following three key modifications to Algorithm 1:

1. Instead of sampling a random point $\mathbf{w}_t$ from the segment (cf. Line 5), we set $\mathbf{w}_t = \mathbf{x}_{t+1}$.
2. To evaluate the alignment at time $t$, we compute both gradients *with the same mini-batch* $\xi_{t+1}$; i.e., we use $\langle \nabla f(\mathbf{x}_{t+1}; \xi_{t+1}), \nabla f(\mathbf{x}_t; \xi_{t+1}) \rangle$ as the first term of the surrogate loss in (5).
3. We omit the clipping step in the learning rate update (7), thus eliminating the hyperparameter $\eta_{\max}$.

Among these modifications, the first and third are mainly for simplicity. In contrast, the second plays a crucial role in the empirical performance of our method, as discussed in the Appendix. Incorporating these changes, the learning rate update rule using FTRL becomes:

$$L_t = \frac{\|\nabla f(\mathbf{x}_{t+1}; \xi_{t+1}) - \nabla f(\mathbf{x}_t; \xi_{t+1})\|}{\|\mathbf{x}_{t+1} - \mathbf{x}_t\|}, \; \eta_{t+1} = \frac{\sum_{s=0}^t \langle \nabla f(\mathbf{x}_{s+1}; \xi_{s+1}), \nabla f(\mathbf{x}_s; \xi_{s+1}) \rangle}{\sum_{s=0}^t L_s \|\mathbf{g}_s(\mathbf{x}_s)\|^2}. \quad (11)$$

**Adam with GALA.** In addition to SGD, we also adapt our GALA to Adam optimizer (Kingma & Ba, 2015). Specifically, the standard Adam update rule is given by

$$\mathbf{m}_t = \beta_1 \mathbf{m}_{t-1} + (1 - \beta_1) \nabla f(\mathbf{x}_t; \xi_t), \quad \mathbf{v}_t = \beta_2 \mathbf{v}_{t-1} + (1 - \beta_2) \nabla f(\mathbf{x}_t; \xi_t)^2,$$

$$\mathbf{d}_t = \frac{\mathbf{m}_t}{\sqrt{\delta + \mathbf{v}_t}}, \quad \mathbf{x}_{t+1} = \mathbf{x}_t - \eta_t \mathbf{d}_t,$$

where all operations are element-wise, and we omit bias correction terms for simplicity. To select the learning rate $\eta_t$ for Adam, we modify the surrogate loss function in (5) as follows: (i) we replace the SGD direction $\nabla f(\mathbf{x}_t; \xi_t)$ with the Adam update direction $\mathbf{d}_t$; (ii) we substitute $\nabla f(\mathbf{w}_t; \xi_t')$ with $\nabla f(\mathbf{x}_t; \xi_t)$, so that the gradient alignment term involves the inner product of stochastic gradients computed on the same mini-batch $\xi_t$; (iii) we estimate $L_t$ using the same heuristic as in (11). These modifications lead to the following learning rate update rule:

$$\eta_{t+1} = \frac{\sum_{s=0}^t \langle \mathbf{d}_s, \nabla f(\mathbf{x}_s; \xi_s) \rangle}{\sum_{s=0}^t L_s \|\mathbf{d}_s\|^2}.$$

*Remark* 8. GALA uses *one sample per iteration*, same as SGD, Adam, and other optimizers. On the other hand, it makes one more gradient call to compute the gradient Lipschitz estimate, similar to AdGD.

## 5.2 RESULTS ON RESNET-18 AND VIT-TINY

**Model and dataset.** We use the `torchvision` implementation of the ResNet-18 model and train it on CIFAR-10, CIFAR-100 and Flower102. We apply random cropping and horizontal flipping as data augmentations during training. For ViT-Tiny, we use the `timm` implementation with pretrained weights and finetune it on Tiny-ImageNet-200. The training uses a richer set of augmentations, including random cropping, horizontal flipping, color jitter, random erasing, and the auto augmentation for ImageNet from `torchvision`.

**Optimizers.** We apply GALA on SGD and Adam, denoted as SGD-GALA and ADAM-GALA, respectively. For ResNet-18 experiments, we compare them against SGD, Adam, as well as parameter-free algorithms such as Mechanic (Cutkosky et al., 2023a), AdGD (Malitsky & Mishchenko, 2020), Prodigy (Mishchenko & Defazio, 2024) and D-Adaptation (Defazio & Mishchenko, 2023) on Adam. For ViT-Tiny, we compared our GALA variants against SGD, AdamW, and Prodigy.

**Setup.** Starting from the same initial model parameters, we run each method with initial learning rates from $[10^{-8}, 1]$ and fix all other parameters at default values, which we report in Appendix E. For ResNet-18, we run all algorithms for 200 epochs with a training batch size of 128, a constant learning rate schedule and zero weight decay. For ViT-Tiny, we train for 100 epochs but increase the

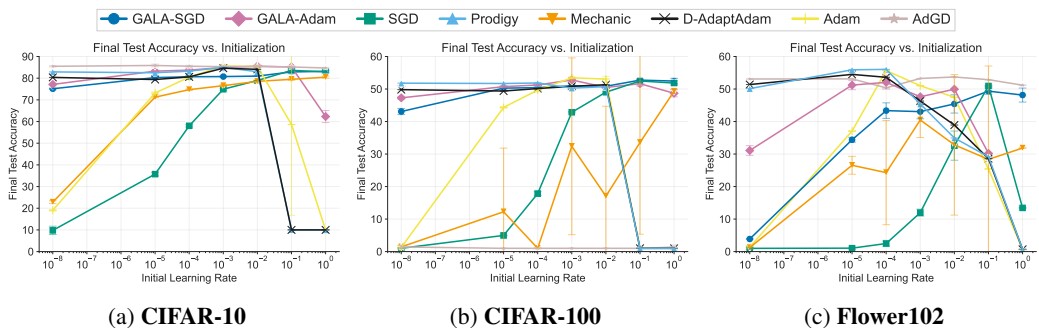

(a) **CIFAR-10**     (b) **CIFAR-100**     (c) **Flower102**

Figure 2: Performance of optimizers in terms of **final test accuracy** for training **ResNet-18** on CIFAR-10, CIFAR-100 and Flower102 from different initial learning rates.

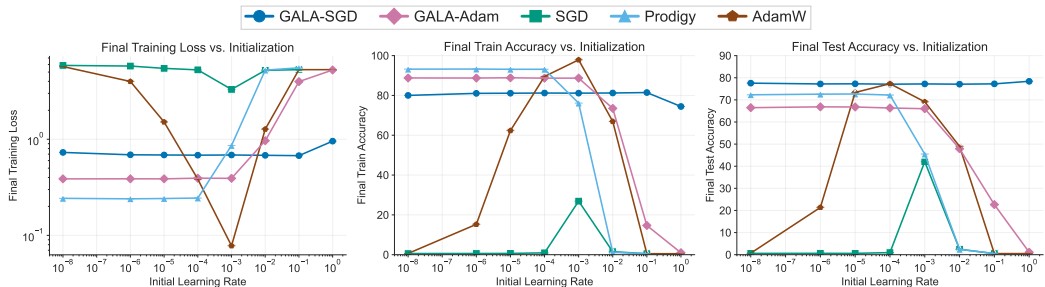

Figure 3: Training loss, training and test accuracy for training **ViT-Tiny** on **Tiny-ImageNet.**

batch size to 1024 to maximize GPU capabilities. We use a cosine schedule with weight decay of 0.05 for AdamW, SGD and Prodigy, while SGD-GALA and ADAM-GALA are run with a constant schedule and zero weight decay.

Due to space limitations, we present a representative subset of our experiments. We provide an extensive empirical study in Section E with more convergence plots and learning rate analysis.

**Results for ResNet-18.** In Figure 2, we show the final test accuracy with respect to initial learning rates when training ResNet-18. SGD-GALA and ADAM-GALA perform consistently across a range of initial learning rates and are competitive with the best-performing method. We validate that GALA enables its base optimizer to achieve competitive performance for a *wider range of initial values, improving stability* for very large/small initializations.

In comparison, AdGD also demonstrates stable performance both for testing (and training). While GALA variants tend to have consistent performance, AdGD has a slight advantage with the test accuracy for CIFAR-10 and Flower102. Note that official AdGD implementation requires an additional mechanism to limit the growth of the learning rate, introducing an extra hyperparameter. We observe that AdGD performs poorly for the CIFAR-100 dataset due to unstable learning rate evolution (see Figure 10c in Section E). Mechanic shows better performance for large initial values, which deteriorates noticeably as the learning rate gets smaller. Both Prodigy and D-Adaptation show one of the best performances when the initial step size is small; however, the performance drops sharply when the initialization is larger than $10^{-3}$.

**Results for ViT-Tiny.** The robustness of GALA is better displayed with the ViT experiments; SGD-GALA improves the performance of SGD from all initializations and shows almost the same performance for all values. ADAM-GALA shows a similar performance improvement particularly for the initialization regimes where AdamW performs poorly, while the best-performing AdamW is better than that of ADAM-GALA. One configuration of AdamW has (one of) the best performance, while this is achieved for a very narrow range of learning rates (from $10^{-4}$ to $10^{-3}$). Prodigy has strong performance for small initializations but the same degradation happens for learning rates larger than $10^{-3}$. Note that for ViT setup, SGD-GALA has *the best* test accuracy *from any initialization*.

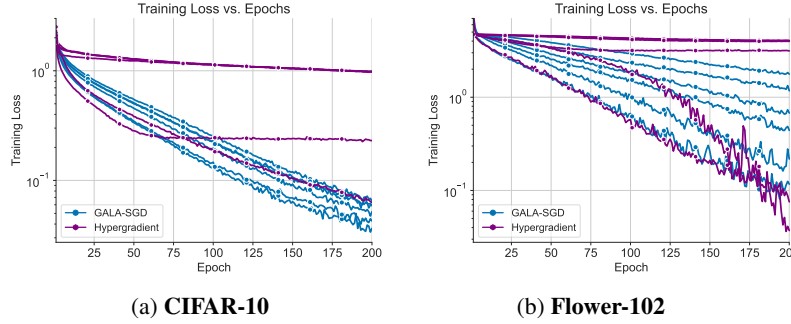

(a) **CIFAR-10**          (b) **Flower-102**

Figure 4: Comparison of **training loss** between SGD-GALA and hypergradient descent from different initial learning rates.

**Comparison with hypergradient descent.** Hypergradient descent method (HGD) does not have convergence guarantees beyond the deterministic, convex setting; however, due to its relevance to our method, we present a comparative empirical study. Unlike our method, HGD comes with an additional hyperparameter: hypergradient learning rate which is used for the gradient update on the *actual learning rate* $\eta_t$. We trained ResNet-18 models on CIFAR-10 and Flower-102 datasets and selected initial value for the actual learning rate $\eta_0$ and the hypergradient step size $\gamma_t$ from the set $\{1, 10^{-1}, 10^{-2}, 10^{-3}, 10^{-4}, 10^{-5}\}$ (a total of 36 runs per dataset). For both datasets, HGD diverged (from all initial learning rates) when the hypergradient learning rate is relatively large ($\{1, 0.1\}$ for CIFAR-10 and 1 for Flower102). The behavior is more stable as the hypergradient step size gets smaller, however, this would impact the convergence speed of the algorithm as the learning rate $\eta_t$ evovles slowly and the decrease in loss value would stagnate for most initializations. To give an advantage to HGD, we picked the best performing hypergradient learning rate for each dataset ($10^{-5}$ for both) and plot the comparison with our method for the same range of step size values $\eta_0$ in Figure 4.

**Evolution of the learning rate.** In Figure 5, we show how the learning rate evolves for ADAM-GALA. When initialized with a large learning rate, GALA tends to reduce it; this might be followed by an increasing regime or a sustained decrease depending on the base value. Conversely, it increases the learning rate when initialized with a small value, which is the ultimate mechanism that enables it to recover from very small initializations. This adaptive behavior arises from the use of an online learning algorithm to select the learning rate, allowing it to adjust dynamically rather than follow a fixed schedule. An interesting observation is that the learning rate for ADAM-GALA, in the majority of cases, initially approaches a value between $10^{-3}$ and $10^{-4}$, which, according to Figure 2, corresponds to the best fixed step

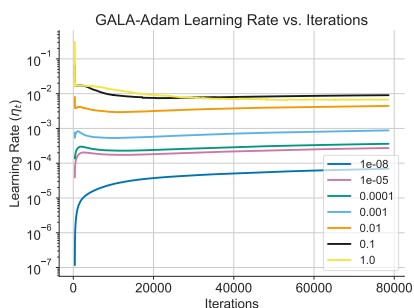

Figure 5: ADAM-GALA's learning rate evolution for training ResNet-18 on CIFAR-100.

size choices when running Adam. As a result of this learning rate adaptation, we observe more consistent convergence paths for methods coupled with GALA.

## 6 CONCLUSION

In this paper, we propose a principled framework, GALA, that dynamically adjusts the learning rate based on gradient alignment and a local curvature estimate. Motivated by convergence analysis, we formulate learning rate selection as a one-dimensional online learning problem and solve it using an online learning algorithm. We establish convergence guarantees for normalized SGD equipped with GALA and conduct preliminary experiments demonstrating that, when combined with SGD or Adam, our method yields robust performance across a wide range of initial learning rates.

One potential limitation of our work is that the convergence analysis is established for one instantiation of GALA and our experiments focus on its integration with SGD and Adam. An interesting future venue is to extend our framework to a broader class of optimizers.

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

APPENDIX

## A  EXTENDED RELATED WORK

**Classical stochastic first-order methods.** Dating back to the seminal work Robbins & Monro (1951), the theoretical behavior of SGD and its many variants have been extensively studied. Considering general smooth functions, it is well-known that the learning rate must decrease at a rate of $\eta_t = O(1/\sqrt{t})$ where $t$ is the iteration counter and also satisfy $\eta_t \leq O(1/L)$. Ghadimi & Lan (2013) established that SGD with a properly chosen learning rate achieves a complexity of $\mathcal{O}(\epsilon^{-2} + \sigma^2 \epsilon^{-4})$, interpolating between deterministic and stochastic rates and matching the lower bounds (Arjevani et al., 2023). However, the choice of the learning rate depends on the problem parameters, i.e., $L, \sigma$, which are typically unknown and prohibitively difficult to estimate in practice. Similar requirements are in place for the learning rate when the objective function is $\rho$-weakly convex (Davis et al., 2020).

**Adaptive and parameter-free optimization methods.** AdaGrad was introduced in two concurrent works McMahan & Streeter (2010); Duchi et al. (2011) for minimizing a sequence of *online* convex losses. The main idea is to compute a time-varying learning rate by accumulating squared norms of stochastic gradients. This fundamental idea paved the way for many algorithms such as Adam (Kingma & Ba, 2015), RMSProp Tieleman & Hinton (2012), Adadelta Zeiler (2012), and their variants, which demonstrate strong empirical performance. Beyond the online optimization setup, they have been shown to automatically adapt to problem-dependent parameters such as smoothness, noise variance, and bounds on gradients. Their convergence properties have been well-studied for the convex setting (Levy, 2017; Levy et al., 2018; Kavis et al., 2019; Joulani et al., 2020; Antonakopoulos et al., 2022; Liu et al., 2023a; Rodomanov et al., 2024) and non-convex setting (Li & Orabona, 2019; Ward et al., 2020; Li & Orabona, 2020; Kavis et al., 2022; Gadat & Gavra, 2022; Faw et al., 2022; Attia & Koren, 2023; Liu et al., 2023b). A downside of the first-generation adaptive methods is the sensitivity to initial learning rate, dampening the practical benefits of their data-adaptive design.

To remedy this, *parameter-free* optimization (Carmon & Hinder, 2022; Ivgi et al., 2023; Khaled et al., 2023; Kreisler et al., 2024; Attia & Koren, 2024) has gained popularity with a focus on augmenting robustness. Essentially, they multiply AdaGrad-type learning rate with a scaling factor that iteratively improves the initial learning rate estimate. Although this helps increase from the initial value, the scaling factor is practically bounded, restricting flexibility. On a related front, a different line of work Malitsky & Mishchenko (2020; 2024); Li & Lan (2024) study parameter-free gradient methods with local curvature estimation for convex, deterministic problems. They are separated from AdaGrad-type methods with non-monotone learning rate that estimates time evolution of local smoothness. A downside to these methods is empirical stability; when the increasing behavior is not tamed properly, optimization performance might be unstable especially for nonconvex problems. Therefore, it is of utmost importance to strike the right balance between flexibility and stability.

**Hypergradient descent.** Originally proposed as a heuristic for stochastic optimization in Almeida et al. (1999), hypergradient descent updates the learning rate by computing the gradient with respect to the learning rate. It was later rediscovered and adapted to modern deep learning Rubio (2017); Baydin et al. (2018), with several subsequent works refining this approach Chandra et al. (2022); Ozkara et al. (2024). Recently, Gao et al. (2024); Chu et al. (2025) provided convergence guarantees from an online learning perspective, though their analysis is limited to deterministic convex settings.

**Online learning-guided methods.** Drawing insights from parameter-free online learning Orabona & Pál (2016), Orabona & Tommasi (2017) reformulate SGD as a coin-betting game and apply a betting algorithm to eliminate the need for a manually tuned learning rate. They also provide convergence guarantees for convex and quasi-convex objectives. Cutkosky et al. (2023a) proposed a general technique for adaptively scaling any base optimization algorithm and learning rate schedule, which is grounded in a black-box reduction framework from parameter-free online learning Cutkosky & Orabona (2018). The work most relevant to ours is that of Zhuang et al. (2019), who consider non-convex stochastic optimization and introduce a surrogate loss technique for selecting the learning rate. However, their method requires knowledge of problem-dependent parameters (e.g., gradient's Lipschitz constant), which limits its flexibility.

# B    PROOF OF LEMMA 1

Recall from (4) that $F(\mathbf{x}_{t+1}) - F(\mathbf{x}_t) = -\eta_t \, \mathbb{E}_{\lambda_t, \xi'_t}[\langle \mathbf{g}'_t(\mathbf{w}_t), \mathbf{g}_t(\mathbf{x}_t)\rangle]$. We now decompose the right-hand side as $\langle \mathbf{g}'_t(\mathbf{w}_t), \mathbf{g}_t(\mathbf{x}_t)\rangle = \langle \mathbf{g}'_t(\mathbf{x}_t), \mathbf{g}_t(\mathbf{x}_t)\rangle + \langle \mathbf{g}'_t(\mathbf{w}_t) - \mathbf{g}'_t(\mathbf{x}_t), \mathbf{g}_t(\mathbf{x}_t)\rangle$, where $\mathbf{g}'_t(\mathbf{x}_t) = \nabla f(\mathbf{x}_t; \xi_t)$. For the first term, since $\xi_t$ and $\xi'$ are independent samples from the distribution $\mathcal{D}$, it holds that $\mathbb{E}[\langle \mathbf{g}'_t(\mathbf{x}_t), \mathbf{g}_t(\mathbf{x}_t)\rangle] = \mathbb{E}[\|\nabla F(\mathbf{x}_t)\|^2]$. Moreover, for the second term, it follows from Cauchy-Schwarz inequality and the definition of $L_t$ that

$$\langle \mathbf{g}'_t(\mathbf{w}_t) - \mathbf{g}'_t(\mathbf{x}_t), \mathbf{g}_t(\mathbf{x}_t)\rangle \geq -\|\mathbf{g}'_t(\mathbf{w}_t) - \mathbf{g}'_t(\mathbf{x}_t)\|\|\mathbf{g}_t(\mathbf{x}_t)\| = -L_t\|\mathbf{w}_t - \mathbf{x}_t\|\|\mathbf{g}_t(\mathbf{x}_t)\|.$$

Since $\mathbf{w}_t = \mathbf{x}_t + \lambda_t(\mathbf{x}_{t+1} - \mathbf{x}_t)$ and $\lambda_t \in [0,1]$, we further have $\|\mathbf{w}_t - \mathbf{x}_t\| = \lambda_t\|\mathbf{x}_{t+1} - \mathbf{x}_t\| \leq \|\mathbf{x}_{t+1} - \mathbf{x}_t\| = \eta_t\|\mathbf{g}_t(\mathbf{x}_t)\|$, which leads to $\langle \mathbf{g}'_t(\mathbf{w}_t) - \mathbf{g}'_t(\mathbf{x}_t), \mathbf{g}_t(\mathbf{x}_t)\rangle \geq -L_t\eta_t\|\mathbf{g}_t(\mathbf{x}_t)\|^2$. By combining both results, we obtain that

$$\mathbb{E}[\langle \mathbf{g}'_t(\mathbf{w}_t), \mathbf{g}_t(\mathbf{x}_t)\rangle] \geq \mathbb{E}[\|\nabla F(\mathbf{x}_t)\|^2] - \mathbb{E}[L_t\eta_t\|\mathbf{g}_t(\mathbf{x}_t)\|^2].$$

Hence, taking expectations on both sides of (4), we can write

$$\begin{aligned}
\mathbb{E}[F(\mathbf{x}_{t+1}) - F(\mathbf{x}_t)] &= -\mathbb{E}[\eta_t\langle \mathbf{g}'_t(\mathbf{w}_t), \mathbf{g}_t(\mathbf{x}_t)\rangle]\\
&= -\mathbb{E}[(\eta_t - \eta)\langle \mathbf{g}'_t(\mathbf{w}_t), \mathbf{g}_t(\mathbf{x}_t)\rangle] - \eta\,\mathbb{E}[\langle \mathbf{g}'_t(\mathbf{w}_t), \mathbf{g}_t(\mathbf{x}_t)\rangle]\\
&\leq -\mathbb{E}[(\eta_t - \eta)\langle \mathbf{g}'_t(\mathbf{w}_t), \mathbf{g}_t(\mathbf{x}_t)\rangle] - \eta\,\mathbb{E}[\|\nabla F(\mathbf{x}_t)\|^2] + \eta\,\mathbb{E}[L_t\eta_t\|\mathbf{g}_t(\mathbf{x}_t)\|^2].
\end{aligned}$$
(12)

Moreover, from Young's inequality $\eta\eta_t \leq \frac{\eta^2}{2} + \frac{\eta_t^2}{2}$, the last term in (12) can be bounded by $L_t\eta\eta_t\|\mathbf{g}_t(\mathbf{x}_t)\|^2 \leq \frac{L_t\|\mathbf{g}_t(\mathbf{x}_t)\|^2\eta^2}{2} + \frac{L_t\|\mathbf{g}_t(\mathbf{x}_t)\|^2\eta_t^2}{2}$. Thus, we obtain

$$\begin{aligned}
\mathbb{E}[F(\mathbf{x}_{t+1}) - F(\mathbf{x}_t)] &\leq -\eta\,\mathbb{E}[\|\nabla F(\mathbf{x}_t)\|^2] - \mathbb{E}[(\eta_t - \eta)\langle \mathbf{g}'_t(\mathbf{w}_t), \mathbf{g}_t(\mathbf{x}_t)\rangle] + \mathbb{E}[L_t\|\mathbf{g}_t(\mathbf{x}_t)\|^2\eta^2]\\
&\quad + \mathbb{E}\left[\frac{L_t\|\mathbf{g}_t(\mathbf{x}_t)\|^2\eta_t^2}{2} - \frac{L_t\|\mathbf{g}_t(\mathbf{x}_t)\|^2\eta^2}{2}\right]\\
&= -\eta\,\mathbb{E}[\|\nabla F(\mathbf{x}_t)\|^2] + \eta^2\,\mathbb{E}[L_t\|\mathbf{g}_t(\mathbf{x}_t)\|^2] + \mathbb{E}[\ell_t(\eta_t) - \ell_t(\eta)],
\end{aligned}$$
(13)

where in the last equality we used the definition of the surrogate loss function in (5). Moreover, Since $L_t \leq L^{\max}$ for any $t \geq 0$ with probability one, we have $\mathbb{E}[L_t\|\mathbf{g}_t(\mathbf{x}_t)\|^2] \leq L^{\max}\,\mathbb{E}[\|\mathbf{g}_t(\mathbf{x}_t)\|^2] \leq L^{\max}(\mathbb{E}[\|\nabla F(\mathbf{x}_t)\|^2] + \sigma^2)$. Plugging this bound in (13) and rearranging, we obtain

$$\mathbb{E}[(\eta - \eta^2 L^{\max})\|\nabla F(\mathbf{x}_t)\|^2] \leq \mathbb{E}[F(\mathbf{x}_t) - F(\mathbf{x}_{t+1})] + L^{\max}\eta^2\sigma^2 + \mathbb{E}[\ell_t(\eta_t) - \ell_t(\eta)].$$

Summing the above inequality from $t = 0$ to $t = T - 1$ yields (6). This completes the proof.

# C    PROOF OF THEOREM 1

We divide the proof of Theorem 1 into the following three steps.

**Step 1:** Following similar arguments as in the proof of Lemma 1, we first bound the function value decrease after one iteration. Its proof can be found in Section C.1.

**Lemma 3.** *For any $\eta > 0$, we have* $\mathbb{E}[F(\mathbf{x}_{t+1}) - F(\mathbf{x}_t)] \leq \mathbb{E}\left[-\frac{\eta}{3}\|\nabla F(\mathbf{x}_t)\| + \max\{L_t, M\}\eta^2 + \frac{8\eta}{3}\|\mathbf{m}_t - \nabla F(\mathbf{x}_t)\|\right] + \mathbb{E}\left[-(\eta_t - \eta)\langle \mathbf{g}'(\mathbf{w}_t), \frac{\mathbf{m}_t}{\|\mathbf{m}_t\|}\rangle + \frac{\max\{L_t, M\}}{2}(\eta_t^2 - \eta^2)\right]$.

In the above bound, the first bracketed term shows up in the analysis of normalized SGD with momentum in Cutkosky & Mehta (2020); it is the upper bound we get when choosing $\eta_t = \eta$. Moreover, the second term in the bracket captures the difference between the actual learning rate $\eta_t$ and the comparator $\eta$. It will be incorporated into the surrogate loss function and be bounded by the regret.

**Step 2:** Next, we controls the approximation error $\mathbb{E}[\|\mathbf{m}_t - \nabla F(\mathbf{x}_t)\|]$ incurred by exponential moving averaging.

**Lemma 4.** *Define $\tilde{L}_t = \frac{\|g'_t(\mathbf{x}_{t+1}) - g'_t(\mathbf{x}_t)\|}{\|\mathbf{x}_{t+1} - \mathbf{x}_t\|}$. Then we have* $\sum_{t=0}^{T-1} \mathbb{E}[\|\mathbf{m}_t - \nabla F(\mathbf{x}_t)\|] \leq \frac{\sigma}{\alpha} + \sigma\sqrt{\alpha}T + \frac{1-\alpha}{\alpha}\sum_{t=0}^{T-2}\mathbb{E}[\tilde{L}_t\eta_t]$.

Lemma 4 upper bounds the approximation error in terms of the learning rate $\eta_t$. As we shall see in the next step, this term will also be incorporated into our surrogate loss and be bounded by the regret.

**Step 3:** By summing the inequality in Lemma 3 from $t = 0$ to $t = T - 1$ and applying Lemma 4, we obtain

$$\mathbb{E}[F(\mathbf{x}_T) - F(\mathbf{x}_0)]$$

$$\leq -\frac{\eta}{3} \mathbb{E}\Big[\sum_{t=0}^{T-1} \|\nabla F(\mathbf{x}_t)\|\Big] + \mathbb{E}\Big[\sum_{t=0}^{T-1} \max\{L_t, M\}\Big]\eta^2 + \frac{8\eta}{3} \mathbb{E}\Big[\sum_{t=0}^{T-1} \|\mathbf{m}_t - \nabla F(\mathbf{x}_t)\|\Big]$$

$$+ \sum_{t=0}^{T-1} \mathbb{E}\Big[-(\eta_t - \eta)\Big\langle \mathbf{g}'(\mathbf{w}_t), \frac{\mathbf{m}_t}{\|\mathbf{m}_t\|}\Big\rangle + \frac{\max\{L_t, M\}}{2}(\eta_t^2 - \eta^2)\Big]$$

$$\leq -\frac{\eta}{3} \mathbb{E}\Big[\sum_{t=0}^{T-1} \|\nabla F(\mathbf{x}_t)\|\Big] + \mathbb{E}\Big[\sum_{t=0}^{T-1} \max\{L_t, M\}\Big]\eta^2 + \frac{8\eta}{3}\Big(\frac{\sigma}{\alpha} + \sigma\sqrt{\alpha}T\Big) + \frac{8(1-\alpha)}{3\alpha} \mathbb{E}\Big[\sum_{t=0}^{T-2} \tilde{L}_t \eta_t \eta\Big]$$

$$+ \sum_{t=0}^{T-1} \mathbb{E}\Big[-(\eta_t - \eta)\Big\langle \mathbf{g}'(\mathbf{w}_t), \frac{\mathbf{m}_t}{\|\mathbf{m}_t\|}\Big\rangle + \frac{\max\{L_t, M\}}{2}(\eta_t^2 - \eta^2)\Big].$$

Moreover, by Young's inequality, we have $\tilde{L}_t \eta_t \eta \leq \frac{\tilde{L}_t}{2}\eta_t^2 + \frac{\tilde{L}_t}{2}\eta^2 = \tilde{L}_t \eta^2 + (\frac{\tilde{L}_t}{2}\eta_t^2 - \frac{\tilde{L}_t}{2}\eta^2)$. Using $\Delta_F = F(\mathbf{x}_0) - F(\mathbf{x}^*) \geq F(\mathbf{x}_0) - F(\mathbf{x}_T)$ and recalling the definition of $\ell_t^{\mathrm{N}}$ in (9), we obtain

$$0 \leq -\frac{\eta}{3} \sum_{t=0}^{T-1} \mathbb{E}[\|\nabla F(\mathbf{x}_t)\|] + \frac{8\eta}{3}\Big(\frac{\sigma}{\alpha} + \sigma\sqrt{\alpha}T\Big) + \mathbb{E}\Big[\sum_{t=0}^{T-1} \max\{L_t, M\} + \frac{8(1-\alpha)\sum_{t=0}^{T-1}\tilde{L}_t}{3\alpha}\Big]\eta^2$$

$$+ \mathbb{E}\Big[\sum_{t=0}^{T-1}(\ell_t^{\mathrm{N}}(\eta_t) - \ell_t^{\mathrm{N}}(\eta))\Big] + \mathbb{E}[\Delta_F].$$

Now for any $\eta \in [0, \eta^{\max}]$, we can upper bound $\sum_{t=0}^{T-1}(\ell_t^{\mathrm{N}}(\eta_t) - \ell_t^{\mathrm{N}}(\eta)) \leq \mathrm{Reg}_T^{\mathrm{N}}$ by definition, and hence we can choose the value of $\eta$ freely from the interval $[0, \eta^{\max}]$ in the above bound. We now consider the following cases:

(i) **Case I:** we have $\sum_{t=0}^{T-1} \mathbb{E}[\|\nabla F(\mathbf{x}_t)\|] \leq 16(\frac{\sigma}{\alpha} + \sigma\sqrt{\alpha}T)$;

(ii) **Case II:** we have $\sum_{t=0}^{T-1} \mathbb{E}[\|\nabla F(\mathbf{x}_t)\|] \geq 16(\frac{\sigma}{\alpha} + \sigma\sqrt{\alpha}T)$. This further implies that

$$0 \leq -\frac{\eta}{6} \sum_{t=0}^{T-1} \mathbb{E}[\|\nabla F(\mathbf{x}_t)\|] + \mathbb{E}\Big[\sum_{t=0}^{T-1} \max\{L_t, M\} + \frac{8(1-\alpha)\sum_{t=0}^{T-1}\tilde{L}_t}{3\alpha}\Big]\eta^2 + \mathbb{E}[\mathrm{Reg}_T^{\mathrm{N}} + \Delta_F]. \tag{14}$$

Moreover, we set the value of $\eta$ as

$$\eta = \min\left\{\frac{\sum_{t=0}^{T-1} \mathbb{E}[\|\nabla F(\mathbf{x}_t)\|]}{\mathbb{E}[12\sum_{t=0}^{T-1} \max\{L_t, M\} + \frac{32(1-\alpha)}{\alpha}\sum_{t=0}^{T-1}\tilde{L}_t]}, \eta^{\max}\right\}. \tag{15}$$

This again leads to two subcases depending on the value of $\eta$:

- If $\eta$ takes the first value in (15), we obtain from (14) that

$$\frac{1}{12}\frac{(\sum_{t=0}^{T-1}\mathbb{E}[\|\nabla F(\mathbf{x}_t)\|])^2}{\mathbb{E}[12\sum_{t=0}^{T-1}\max\{L_t, M\} + \frac{32(1-\alpha)}{\alpha}\sum_{t=0}^{T-1}\tilde{L}_t]} \leq \mathbb{E}[\mathrm{Reg}_T^{\mathrm{N}} + \Delta_F].$$

To simplify the notation, let $M = \Delta_F + \mathrm{Reg}_T^{\mathrm{N}}$. With some algebraic manipulation and using the fact that $\sqrt{a+b} \leq \sqrt{a} + \sqrt{b}$, we obtain

$$\sum_{t=0}^{T-1} \mathbb{E}[\|\nabla F(\mathbf{x}_t)\|] \leq 12\sqrt{\mathbb{E}[M]\,\mathbb{E}\Big[\sum_{t=0}^{T-1} \max\{L_t, M\}\Big]} + 8\sqrt{\frac{6(1-\alpha)}{\alpha}}\sqrt{\mathbb{E}[M]\,\mathbb{E}\Big[\sum_{t=0}^{T-1}\tilde{L}_t\Big]}.$$

- If $\eta$ takes the second value in (15), then $\eta = \eta^{\max} \leq$ $\frac{\sum_{t=0}^{T-1} \mathbb{E}[\|\nabla F(\mathbf{x}_t)\|]}{\mathbb{E}[12\sum_{t=0}^{T-1}\max\{L_t,M\}+\frac{32(1-\alpha)}{\alpha}\sum_{t=0}^{T-1}\tilde{L}_t]}$. In this case, we obtain from (14) that

$$\frac{\eta^{\max}}{12}\sum_{t=0}^{T-1}\mathbb{E}[\|\nabla F(\mathbf{x}_t)\|] \leq \mathbb{E}[M] \quad \Rightarrow \quad \sum_{t=0}^{T-1}\mathbb{E}[\|\nabla F(\mathbf{x}_t)\|] \leq \frac{12\,\mathbb{E}[M]}{\eta^{\max}}.$$

Combining the upper bounds in all cases and using the definition of $L_T^{\text{avg}}$, we can deduce that

$$\sum_{t=0}^{T-1}\mathbb{E}[\|\nabla F(\mathbf{x}_t)\|] \leq 16(\frac{\sigma}{\alpha}+\sigma\sqrt{\alpha}T)+12\sqrt{\mathbb{E}[M]L_T^{\text{avg}}T}+8\sqrt{\frac{6(1-\alpha)}{\alpha}}\sqrt{\mathbb{E}[M]L_T^{\text{avg}}T}+\frac{12\,\mathbb{E}[M]}{\eta^{\max}}.$$
(16)

Finally, we can choose the parameter $\alpha$ to optimize the above upper bound. Specifically, we let

$$\alpha = \min\Big\{\frac{1}{\sigma\sqrt{T}},1\Big\}.$$

If $\frac{1}{\sigma\sqrt{T}} \leq 1$, then we have $\frac{16\sigma}{\alpha} \leq 16\sigma^2\sqrt{T}$, $16\sigma\sqrt{\alpha}T \leq 16\sigma^{1/2}T^{3/4}$, and $8\sqrt{\frac{6(1-\alpha)}{\alpha}}\sqrt{\mathbb{E}[M]L_T^{\text{avg}}T} \leq 8\sqrt{6}\sigma^{1/2}(L_T^{\text{avg}}\mathbb{E}[M])^{1/2}T^{3/4}$. Otherwise, if $\frac{1}{\sigma\sqrt{T}} > 1$, then $\alpha = 1$ and we have $\frac{16\sigma}{\alpha} = 16\sigma \leq \frac{16}{\sqrt{T}}$, $16\sigma\sqrt{\alpha}T \leq 16\sigma T \leq 16\sqrt{T}$, and $8\sqrt{\frac{6(1-\alpha)}{\alpha}}\sqrt{\mathbb{E}[M]L_T^{\text{avg}}T} = 0$. Hence, combining both cases, we conclude that

$$16(\frac{\sigma}{\alpha}+\sigma\sqrt{\alpha}T)+8\sqrt{\frac{6(1-\alpha)}{\alpha}}\sqrt{\mathbb{E}[M]L_T^{\text{avg}}T}$$
$$\leq 16(\sigma^2+2)\sqrt{T}+(16+8\sqrt{6}(L_T^{\text{avg}}\mathbb{E}[M])^{1/2})\sigma^{1/2}T^{3/4}.$$

By using the above bound and dividing both sides by $T$ in (16), we arrive at

$$\frac{1}{T}\sum_{t=0}^{T-1}\mathbb{E}[\|\nabla F(\mathbf{x}_t)\|] \leq \frac{(16+8\sqrt{6}(L_T^{\text{avg}}\mathbb{E}[M])^{1/2})\sigma^{1/2}}{T^{1/4}}+\frac{16(\sigma^2+2)}{\sqrt{T}}+12\frac{\sqrt{L_T^{\text{avg}}\mathbb{E}[M]}}{\sqrt{T}}$$
$$+\frac{12\,\mathbb{E}[M]}{\eta^{\max}T}.$$

This completes the proof of Theorem 1.

### C.1 PROOF OF LEMMA 3

Similar to the arguments in Section 3, we first apply the fundamental theorem of calculus to get $F(\mathbf{x}_{t+1})-F(\mathbf{x}_t) = \langle\boldsymbol{\nabla}_t,\mathbf{x}_{t+1}-\mathbf{x}_t\rangle = -\eta_t\langle\boldsymbol{\nabla}_t,\frac{\mathbf{m}_t}{\|\mathbf{m}_t\|}\rangle$. Since $\boldsymbol{\nabla}_t = \mathbb{E}_{\lambda_t}[\nabla F(\mathbf{w}_t)] = \mathbb{E}_{\lambda_t,\xi_t'}[\mathbf{g}_t'(\mathbf{w}_t)]$, we further have

$$F(\mathbf{x}_{t+1})-F(\mathbf{x}_t) = -\eta_t\mathop{\mathbb{E}}_{\lambda_t,\xi_t'}\Big[\Big\langle\mathbf{g}_t'(\mathbf{w}_t),\frac{\mathbf{m}_t}{\|\mathbf{m}_t\|}\Big\rangle\Big].$$
(17)

Next, we decompose the right-hand side of (17) as

$$\Big\langle\mathbf{g}_t'(\mathbf{w}_t),\frac{\mathbf{m}_t}{\|\mathbf{m}_t\|}\Big\rangle = \Big\langle\mathbf{g}_t'(\mathbf{x}_t),\frac{\mathbf{m}_t}{\|\mathbf{m}_t\|}\Big\rangle + \Big\langle\mathbf{g}_t'(\mathbf{w}_t)-\mathbf{g}_t'(\mathbf{x}_t),\frac{\mathbf{m}_t}{\|\mathbf{m}_t\|}\Big\rangle$$
$$\geq \Big\langle\mathbf{g}_t'(\mathbf{x}_t),\frac{\mathbf{m}_t}{\|\mathbf{m}_t\|}\Big\rangle - \|\mathbf{g}_t'(\mathbf{w}_t)-\mathbf{g}_t'(\mathbf{x}_t)\|,$$

where we used Cauchy-Schwarz inequality in the last step. Using the definition of $L_t$, we have

$$\|\mathbf{g}_t'(\mathbf{w}_t)-\mathbf{g}_t'(\mathbf{x}_t)\| \leq L_t\|\mathbf{w}_t-\mathbf{x}_t\| = L_t\lambda_t\|\mathbf{x}_{t+1}-\mathbf{x}_t\| \leq L_t\eta_t \leq \max\{L_t,M\}\eta_t.$$
(18)

Moreover, since $\mathbf{g}_t'(\mathbf{x}_t)$ and $\mathbf{m}_t$ are independent conditioned on $\mathbf{x}_t$, we further have $\mathbb{E}[\langle\mathbf{g}_t'(\mathbf{x}_t),\frac{\mathbf{m}_t}{\|\mathbf{m}_t\|}\rangle] = \mathbb{E}[\langle\nabla F(\mathbf{x}_t),\frac{\mathbf{m}_t}{\|\mathbf{m}_t\|}\rangle]$, which is further lower bounded in the following lemma.

**Lemma 5.** *We have* $\langle\nabla F(\mathbf{x}_t),\frac{\mathbf{m}_t}{\|\mathbf{m}_t\|}\rangle \geq \frac{1}{3}\|\nabla F(\mathbf{x}_t)\| - \frac{8}{3}\|\mathbf{m}_t-\nabla F(\mathbf{x}_t)\|$.

*Proof.* Our proof is inspired by (Cutkosky & Mehta, 2020, Lemma 2). We consider two cases:

(i) If $\|\mathbf{m}_t - \nabla F(\mathbf{x}_t)\| \leq \frac{1}{2}\|\nabla F(\mathbf{x}_t)\|$, then by the triangle inequality, we have $\|\mathbf{m}_t\| \leq \frac{3}{2}\|\nabla F(\mathbf{x}_t)\|$. Therefore, we have

$$
\frac{1}{\|\mathbf{m}_t\|}\langle \nabla F(\mathbf{x}_t), \mathbf{m}_t \rangle = \frac{1}{\|\mathbf{m}_t\|}(\|\nabla F(\mathbf{x}_t)\|^2 + \langle \nabla F(\mathbf{x}_t), \mathbf{m}_t - \nabla F(\mathbf{x}_t) \rangle)
$$

$$
\geq \frac{1}{\|\mathbf{m}_t\|}(\|\nabla F(\mathbf{x}_t)\|^2 - \frac{1}{2}\|\nabla F(\mathbf{x}_t)\|^2) \geq \frac{1}{3}\|\nabla F(\mathbf{x}_t)\|,
$$

where we used $\|\mathbf{m}_t - \nabla F(\mathbf{x}_t)\| \leq \frac{1}{2}\|\nabla F(\mathbf{x}_t)\|$ in the first inequality and $\|\mathbf{m}_t\| \leq \frac{3}{2}\|\nabla F(\mathbf{x}_t)\|$ in the second one. Since $\frac{8}{3}\|\mathbf{m}_t - \nabla F(\mathbf{x}_t)\| \geq 0$, the result in Lemma 5 holds under this case.

(ii) Otherwise, if $\|\mathbf{m}_t - \nabla F(\mathbf{x}_t)\| > \frac{1}{2}\|\nabla F(\mathbf{x}_t)\|$, we can instead use Cauchy-Schwarz inequality to bound

$$
\frac{1}{\|\mathbf{m}_t\|}\langle \nabla F(\mathbf{x}_t), \mathbf{m}_t \rangle \geq -\|\nabla F(\mathbf{x}_t)\| = \frac{1}{3}\|\nabla F(\mathbf{x}_t)\| - \frac{4}{3}\|\nabla F(\mathbf{x}_t)\|
$$

$$
\geq \frac{1}{3}\|\nabla F(\mathbf{x}_t)\| - \frac{8}{3}\|\mathbf{m}_t - \nabla F(\mathbf{x}_t)\|,
$$

where we used $\|\mathbf{m}_t - \nabla F(\mathbf{x}_t)\| > \frac{1}{2}\|\nabla F(\mathbf{x}_t)\|$ in the last inequality.

This completes the proof. $\qquad\square$

Combining (18) and Lemma 5, we obtain that

$$
\mathbb{E}\Big[\Big\langle \mathbf{g}'_t(\mathbf{w}_t), \frac{\mathbf{m}_t}{\|\mathbf{m}_t\|}\Big\rangle\Big] \geq \mathbb{E}\Big[\frac{1}{3}\|\nabla F(\mathbf{x}_t)\| - \frac{8}{3}\|\mathbf{m}_t - \nabla F(\mathbf{x}_t)\| - \max\{L_t, M\}\eta_t\Big].
$$

Hence, it further follows from (17) that

$$
\mathbb{E}[F(\mathbf{x}_{t+1}) - F(\mathbf{x}_t)] = -\mathbb{E}\Big[(\eta_t - \eta)\Big\langle \mathbf{g}'_t(\mathbf{w}_t), \frac{\mathbf{m}_t}{\|\mathbf{m}_t\|}\Big\rangle\Big] - \eta\,\mathbb{E}\Big[\Big\langle \mathbf{g}'_t(\mathbf{w}_t), \frac{\mathbf{m}_t}{\|\mathbf{m}_t\|}\Big\rangle\Big]
$$

$$
\leq -\mathbb{E}\Big[(\eta_t - \eta)\Big\langle \mathbf{g}'_t(\mathbf{w}_t), \frac{\mathbf{m}_t}{\|\mathbf{m}_t\|}\Big\rangle - \frac{\eta\|\nabla F(\mathbf{x}_t)\|}{3} + \frac{8\eta\|\mathbf{m}_t - \nabla F(\mathbf{x}_t)\|}{3}
$$

$$
+ \max\{L_t, M\}\eta_t\eta\Big].
$$

By using Young's inequality $\eta_t\eta \leq \frac{\eta_t^2}{2} + \frac{\eta^2}{2}$ and rearranging, we obtain the inequality in Lemma 3.

## C.2 PROOF OF LEMMA 4

From the update rule in (8), we can write

$$
\mathbf{m}_t - \nabla F(\mathbf{x}_t) = (1 - \alpha)(\mathbf{m}_{t-1} - \nabla F(\mathbf{x}_{t-1})) + \alpha(\nabla f(\mathbf{x}_t; \xi_t) - \nabla F(\mathbf{x}_t))
$$
$$
+ (1 - \alpha)(\nabla F(\mathbf{x}_{t-1}) - \nabla F(\mathbf{x}_t)). \tag{19}
$$

Define the stochastic gradient error $\mathbf{e}_t = \nabla f(\mathbf{x}_t; \xi_t) - \nabla F(\mathbf{x}_t)$. By Assumption 2, we have $\mathbb{E}[\mathbf{e}_t] = 0$ and $\mathbb{E}[\|\mathbf{e}_t\|^2] \leq \sigma^2$. Moreover, by multiplying both sides of (19) with $(1 - \alpha)^{-t}$, we have

$$
(\mathbf{m}_t - \nabla F(\mathbf{x}_t))(1 - \alpha)^{-t} = (\mathbf{m}_{t-1} - \nabla F(\mathbf{x}_{t-1}))(1 - \alpha)^{-t+1} + \alpha\mathbf{e}_t(1 - \alpha)^{-t}
$$
$$
+ (\nabla F(\mathbf{x}_{t-1}) - \nabla F(\mathbf{x}_t))(1 - \alpha)^{-t+1}.
$$

Note that we set $\mathbf{m}_0 = \nabla f(\mathbf{x}_0; \xi_0)$. Thus, by summing the above inequality, we obtain

$$
(\mathbf{m}_t - \nabla F(\mathbf{x}_t))(1 - \alpha)^{-t} = \mathbf{e}_0 + \sum_{s=1}^{t} \mathbf{e}_s\alpha(1 - \alpha)^{-s} + \sum_{s=1}^{t} (\nabla F(\mathbf{x}_{s-1}) - \nabla F(\mathbf{x}_s))(1 - \alpha)^{-s+1}.
$$

Therefore, it follows from the triangle inequality that

$$\|\mathbf{m}_t - \nabla F(\mathbf{x}_t)\| \le \|\mathbf{e}_0\|(1-\alpha)^t + \left\|\sum_{s=1}^{t} \mathbf{e}_s \alpha(1-\alpha)^{t-s}\right\| + \sum_{s=1}^{t} \|\nabla F(\mathbf{x}_{s-1}) - \nabla F(\mathbf{x}_s)\|(1-\alpha)^{t-s+1}.$$
(20)

By Jensen's inequality and the fact that $\{\xi_s\}_{s=1}^{t}$ are i.i.d. sampled from $\mathcal{D}$, we have $\mathbb{E}[\|\mathbf{e}_0\|] \le \sqrt{\mathbb{E}[\|\mathbf{e}_0\|^2]} = \sigma$ and

$$\mathbb{E}\left\|\sum_{s=1}^{t} \mathbf{e}_s \alpha(1-\alpha)^{t-s}\right\| \le \sqrt{\mathbb{E}\left\|\sum_{s=1}^{t} \mathbf{e}_s \alpha(1-\alpha)^{t-s}\right\|^2} \le \sqrt{\sum_{s=1}^{t} \sigma^2 \alpha^2 (1-\alpha)^{2(t-s)}}.$$

Moreover, it also follows from Jensen's inequality that $\mathbb{E}[\|\nabla F(\mathbf{x}_{s-1}) - \nabla F(\mathbf{x}_s)\|] \le \mathbb{E}[\|\nabla f(\mathbf{x}_s; \xi_s) - \nabla f(\mathbf{x}_{s-1}; \xi_s)\|] = \tilde{L}_{s-1}\|\mathbf{x}_s - \mathbf{x}_{s-1}\| = \tilde{L}_{s-1}\eta_{s-1}$. Hence, by taking the expectation on both sides of (20), we further have

$$\mathbb{E}[\|\mathbf{m}_t - \nabla F(\mathbf{x}_t)\|] \le \sigma(1-\alpha)^t + \sigma\alpha\sqrt{\sum_{s=1}^{t}(1-\alpha)^{2(t-s)}} + \sum_{s=1}^{t} \mathbb{E}[\tilde{L}_{s-1}\eta_{s-1}](1-\alpha)^{t-s+1}$$

$$\le \sigma(1-\alpha)^t + \sigma\alpha\sqrt{\frac{1}{1-(1-\alpha)^2}} + \sum_{s=0}^{t-1} \mathbb{E}[\tilde{L}_s\eta_s](1-\alpha)^{t-s}$$

$$\le \sigma(1-\alpha)^t + \sigma\sqrt{\alpha} + \sum_{s=0}^{t-1} \mathbb{E}[\tilde{L}_s\eta_s](1-\alpha)^{t-s}.$$

By summing the above inequality from $t=0$ to $t = T-1$, we obtain that

$$\sum_{t=0}^{T-1} \mathbb{E}[\|\mathbf{m}_t - \nabla F(\mathbf{x}_t)\|] \le \sigma\sum_{t=0}^{T-1}(1-\alpha)^t + \sigma\sqrt{\alpha}T + \sum_{t=0}^{T-1}\sum_{s=0}^{t-1} \mathbb{E}[\tilde{L}_s\eta_s](1-\alpha)^{t-s}$$

Since $\sum_{t=0}^{T-1}(1-\alpha)^t \le \frac{1}{\alpha}$ and $\sum_{t=0}^{T-1}\sum_{s=0}^{t-1} \mathbb{E}[\tilde{L}_s\eta_s](1-\alpha)^{t-s} = \sum_{s=0}^{T-2}\sum_{t=s+1}^{T-1} \mathbb{E}[\tilde{L}_s\eta_s](1-\alpha)^{t-s} \le \frac{1-\alpha}{\alpha}\sum_{s=0}^{T-2} \mathbb{E}[\tilde{L}_s\eta_s]$, we obtain Lemma 4.

## D  PROOF OF LEMMA 2

As discussed in Section 4, our update for $\eta$ can be viewed as an instance of the optimistic FTRL algorithm. Therefore, we can invoke the convergence bound in (Orabona, 2019, Theorem 7.39), where $\psi_1 = \cdots = \psi_T = \frac{\delta}{2}\eta^2$ and $\tilde{\ell}_{t+1}(\eta) = -\eta\langle \mathbf{g}_{t+1}(\mathbf{x}_{t+1}), \frac{\mathbf{m}_{t+1}}{\|\mathbf{m}_{t+1}\|}\rangle$. Moreover, note that $\frac{\delta}{2}\eta^2 + \sum_{s=0}^{t} \ell_s^N(\eta)$ is $(\delta + \sum_{s=0}^{t}(\max\{L_s, M\} + \frac{8(1-\alpha)}{3\alpha}\tilde{L}_s))$-strongly convex, and $|(\ell_t^N)'(\eta_t) - \tilde{\ell}_t'(\eta_t)| = |-\langle \mathbf{g}_t'(\mathbf{w}_t) - \mathbf{g}_t(\mathbf{x}_t), \frac{\mathbf{m}_t}{\|\mathbf{m}_t\|}\rangle + \max\{L_t, M\}\eta_t + \frac{8(1-\alpha)}{3\alpha}\tilde{L}_t\eta_t| \le \|\mathbf{g}_t'(\mathbf{w}_t) - \mathbf{g}_t(\mathbf{x}_t)\| + \max\{L_t, M\}\eta_t + \frac{8(1-\alpha)}{3\alpha}\tilde{L}_t\eta_t$. Hence, we have

$$\sum_{t=0}^{T-1}(\ell_t(\eta_t) - \ell_t(\eta)) \le \frac{\delta}{2}\eta^2 + \sum_{t=0}^{T-1} \frac{(\max\{L_t, M\}\eta_t + \frac{8(1-\alpha)}{3\alpha}\tilde{L}_t\eta_t + \|\mathbf{g}_t'(\mathbf{w}_t) - \mathbf{g}_t(\mathbf{x}_t)\|)^2}{2\delta + 2\sum_{s=0}^{t}(\max\{L_s, M\} + \frac{8(1-\alpha)\tilde{L}_s}{3\alpha})}. \quad (21)$$

Moreover, by the triangle inequality and the definition of $L_t$, we have $\|\mathbf{g}_t'(\mathbf{w}_t) - \mathbf{g}_t(\mathbf{x}_t)\| = \|\nabla f(\mathbf{w}_t; \xi_t') - \nabla f(\mathbf{x}_t; \xi_t)\| \le \|\nabla f(\mathbf{w}_t; \xi_t') - \nabla f(\mathbf{x}_t; \xi_t')\| + \|\nabla f(\mathbf{x}_t; \xi_t') - \nabla f(\mathbf{x}_t; \xi_t)\| \le L_t\eta_t + \|\nabla f(\mathbf{x}_t; \xi_t') - \nabla f(\mathbf{x}_t; \xi_t)\|$. So we can further bound the summand in (21) by

$$\frac{(2\max\{L_t, M\}\eta_t + \frac{8(1-\alpha)}{3\alpha}\tilde{L}_t\eta_t + \|\nabla f(\mathbf{x}_t; \xi_t') - \nabla f(\mathbf{x}_t; \xi_t)\|)^2}{2\delta + 2\sum_{s=0}^{t}(\max\{L_s, M\} + \frac{8(1-\alpha)\tilde{L}_s}{3\alpha})}$$

$$\le \frac{\eta_t^2(2\max\{L_t, M\} + \frac{8(1-\alpha)}{3\alpha}\tilde{L}_t)^2 + \|\nabla f(\mathbf{x}_t; \xi_t') - \nabla f(\mathbf{x}_t; \xi_t)\|^2}{\delta + \sum_{s=0}^{t}(\max\{L_s, M\} + \frac{8(1-\alpha)\tilde{L}_s}{3\alpha})}.$$

In the following, we will upper bound the two sums

$$\sum_{t=0}^{T-1} \frac{\eta_t^2(2\max\{L_t, M\} + \frac{8(1-\alpha)}{3\alpha}\tilde{L}_t)^2}{\delta + \sum_{s=0}^{t}(\max\{L_s, M\} + \frac{8(1-\alpha)\tilde{L}_s}{3\alpha})} \quad \text{and} \quad \sum_{t=0}^{T-1} \frac{\|\nabla f(\mathbf{x}_t; \xi_t') - \nabla f(\mathbf{x}_t; \xi_t)\|^2}{\delta + \sum_{s=0}^{t}(\max\{L_s, M\} + \frac{8(1-\alpha)\tilde{L}_s}{3\alpha})}$$

separately.

By our assumption, $\max\{L_t, \tilde{L}_t\} \leq L^{\max}$ with probability one and $\eta_t \leq \eta^{\max}$. Thus, we can derive

$$\sum_{t=0}^{T-1} \frac{\eta_t^2(2\max\{L_t, M\} + \frac{8(1-\alpha)}{3\alpha}\tilde{L}_t)^2}{\delta + \sum_{s=0}^{t}(\max\{L_s, M\} + \frac{8(1-\alpha)\tilde{L}_s}{3\alpha})} \tag{22}$$

$$\leq \frac{28(\eta^{\max})^2\max\{L^{\max}, M\}}{3\alpha} \sum_{t=0}^{T-1} \frac{\max\{L_t, M\} + \frac{8(1-\alpha)\tilde{L}_t}{3\alpha}}{\delta + \sum_{s=0}^{t}(\max\{L_s, M\} + \frac{8(1-\alpha)\tilde{L}_s}{3\alpha})}. \tag{23}$$

Now we can apply the following lemma.

**Lemma 6.** *For any nonnegative sequence $\{a_t\}_{t=0}^{T-1}$ and $\delta > 0$, it holds that $\sum_{t=0}^{T-1} \frac{a_t}{\delta + \sum_{s=0}^{t} a_s} \leq \log\left(1 + \frac{\sum_{t=0}^{T-1} a_t}{\delta}\right)$.*

*Proof.* For any $t \geq 0$, we have $\frac{a_t}{\delta + \sum_{s=0}^{t} a_s} = 1 - \frac{\delta + \sum_{s=0}^{t-1} a_s}{\delta + \sum_{s=0}^{t} a_s} \leq \log(\frac{\delta + \sum_{s=0}^{t} a_s}{\delta + \sum_{s=0}^{t-1} a_s})$, where we used the fact that $1 - x \leq \log(\frac{1}{x})$ for any $x \geq 0$. Hence, by summing the inequality from $t = 0$ to $t = T-1$ we obtain $\sum_{t=0}^{T-1} \frac{a_t}{\delta + \sum_{s=0}^{t} a_s} \leq \log(\frac{\delta + \sum_{t=0}^{T-1} a_t}{\delta}) = \log(1 + \frac{\sum_{t=0}^{T-1} a_t}{\delta})$. $\square$

Hence, by applying Lemma 6 to (22), we get

$$\sum_{t=0}^{T-1} \frac{\eta_t^2(2\max\{L_t, M\} + \frac{8(1-\alpha)}{3\alpha}\tilde{L}_t)^2}{\delta + \sum_{s=0}^{t}(\max\{L_s, M\} + \frac{8(1-\alpha)\tilde{L}_s}{3\alpha})}$$

$$\leq \frac{28(\eta^{\max})^2\max\{L^{\max}, M\}}{3\alpha} \log\left(1 + \frac{\sum_{t=0}^{T-1}(\max\{L_t, M\} + \frac{8(1-\alpha)\tilde{L}_t}{3\alpha})}{\delta}\right)$$

$$\leq \frac{28(\eta^{\max})^2\max\{L^{\max}, M\}}{3\alpha} \log\left(1 + \frac{11\max\{L^{\max}, M\}}{3\alpha\delta}T\right).$$

By our choice of $\eta^{\max} = \sqrt{\alpha}\bar{\eta}$, it becomes $\mathcal{O}\left(\bar{\eta}^2\max\{L^{\max}, M\} \log\left(1 + \frac{\max\{L^{\max}, M\}}{\alpha\delta}T\right)\right)$. For the second term, since $\frac{1}{t+1}\sum_{s=0}^{t} \max\{L_s, M\} \geq M$, using Assumption 2, we have

$$\mathbb{E}\left[\sum_{t=0}^{T-1} \frac{\|\nabla f(\mathbf{x}_t; \xi_t') - \nabla f(\mathbf{x}_t; \xi_t)\|^2}{\delta + \sum_{s=0}^{t}(\max\{L_s, M\} + \frac{8(1-\alpha)\tilde{L}_s}{3\alpha})}\right] \leq \sum_{t=0}^{T-1} \frac{\mathbb{E}[\|\nabla f(\mathbf{x}_t; \xi_t') - \nabla f(\mathbf{x}_t; \xi_t)\|^2]}{M(t+1)}$$

$$= \sum_{t=0}^{T-1} \frac{2\sigma^2}{M(t+1)} \leq \frac{2\sigma^2}{M}(1 + \log(T)).$$

Lemma 2 now follows from combining the above two bounds.

# E ADDITIONAL EXPERIMENTS AND DETAILS

In this section, we discuss additional implementation details, present the seeded runs for the experiments in the main text, and include additional results on two other datasets from torchvision.

## E.1 IMPLEMENTATION DETAILS

**Hardware** Our experiments were conducted on a cluster with NVIDIA A100 GPUs (96GB memory) and 120GB system RAM. The CIFAR-10 experiments with multiple random seeds required approximately 96 GPU hours, and both the CIFAR-100 and the Flower102 experiments required approximately 192 GPU hours.

**Additional hyperparameters** Since our main focus is on adaptive learning rate selection, we set all other hyperparameters to their standard default values. Specifically, for ADAM-GALA, we fix $\beta_1 = 0.9$, $\beta_2 = 0.999$, and $\delta = 10^{-8}$, consistent with the settings used for Adam and AdamW. For SGD with momentum, we set the momentum parameter to $0.9$. While training ViT-Tiny, we set the weight decay for AdamW and SGD to be $0.05$, and we kept it $0$ for SGD-GALA and ADAM-GALA.

Resnet-18 experiments use a constant learning rate schedule for all algorithms. We observe that a popular schedule, such as cosine, does not change the relative behavior and hence we report the results for a constant learning rate schedule. For the ViT-Tiny experiment, we follow the standard in the literature and run AdamW and SGD with cosine schedules, while we used a constant schedule for our GALA variants to demonstrate the learning rate adaptation without external intervention.

**Mechanic** The Mechanic algorithm, proposed in Cutkosky et al. (2023a), provides a general framework for adaptively selecting the learning rate of any base optimizer. At each iteration, it proceeds as follows:

- Sample $\xi_t \sim \mathcal{D}$ and compute the stochastic gradient $\mathbf{g}_t = \nabla f(\mathbf{x}_t; \xi_t)$;
- Use $\mathbf{g}_t$ to compute the update direction $\mathbf{u}_t$ via the base optimizer and update the cumulative direction $\mathbf{\Delta}_{t+1} = \mathbf{\Delta}_t + \mathbf{u}_t$;
- An internal online learner selects a learning rate $s_{t+1}$;
- Update the iterate: $\mathbf{x}_{t+1} = \mathbf{x}_1 + s_{t+1}\mathbf{\Delta}_{t+1}$.

For example, if the base optimizer is SGD, then $\mathbf{u}_t = -\eta\mathbf{g}_t$. In our experiments, we apply Mechanic to both SGD and its momentum variant and vary the initial learning rate $\eta$, using the official implementation available at `https://github.com/optimizedlearning/mechanic`. Since the performance of standard and momentum versions were comparable, we only include the Mechanic on SGD without moment in order to preserve readability of the plots.

**AdGD** The update rule of AdGD in Malitsky & Mishchenko (2020) for the stochastic setting is given by

$$\eta_t = \min\left\{\sqrt{1 + \alpha\frac{\eta_{t-1}}{\eta_{t-2}}}\eta_{t-1}, \frac{\|\mathbf{x}_t - \mathbf{x}_{t-1}\|}{2\|\nabla f(\mathbf{x}_t; \xi_t) - \nabla f(\mathbf{x}_{t-1}; \xi_t)\|}\right\},$$

$$\mathbf{x}_{t+1} = \mathbf{x}_t - \eta_t\nabla f(\mathbf{x}_t; \xi_t),$$

where $\alpha = 1$ in the original algorithm, which is analyzed under deterministic gradients. In practice, the authors recommend using smaller values of $\alpha$ to improve stability and avoid spikes in the loss curve. For example, they report that for ResNet-18 on CIFAR-10, setting $\alpha = 0.02$ yields the best performance. Following their recommendation, we use this value in all of our experiments.

**D-Adaptation and Prodigy** Among the class of *parameter-free* methods, D-Adaptation (Defazio & Mishchenko, 2023) and Prodigy (Mishchenko & Defazio, 2024) are considered to be state-of-the-art with 3 million downloads each on GitHub. The goal of parameter-free optimization is to eliminate the dependence on the initial learning rate by approximating the initial distance to the solution. Essentially, the step size is based on the AdaGrad step size with a *certificate* for initial distance, i.e., $\|x_0 - x^*\|$, that iteratively improves every step. Note that the algorithm still requires an initial value for the certificate, $d_0$, which is equivalent to the initial learning rate. We keep all the other parameters of both algorithms the same as in the GitHub implementation and the paper. We downloaded and installed the official algorithm packages from `https://github.com/facebookresearch/dadaptation` for D-Adaptation and `https://github.com/konstmish/prodigy` for Prodigy.

### E.2 EXPERIMENTS

#### E.2.1 RESNET-18

Alongside CIFAR-10, we ran experiments with two more datasets, CIFAR-100 Krizhevsky (2009) and Oxford 102 Flower dataset (Flower102) Nilsback & Zisserman (2008), on ResNet-18. In order to quantify the error due to randomness, we ran the experiments with three different random seeds, which we report as error bars. Figure 6, Figure 7, and Figure 8 report the final training loss, training accuracy, and testing accuracy with respect to different learning rates, respectively.

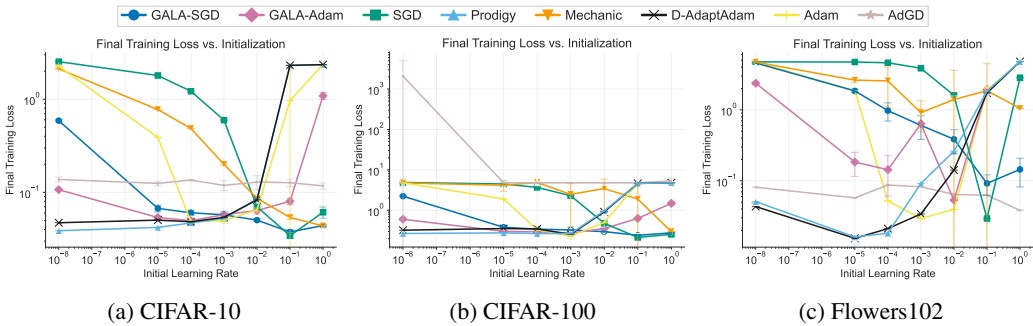

(a) CIFAR-10      (b) CIFAR-100      (c) Flowers102

Figure 6: **Final training loss** obtained from different initial learning rates for CIFAR-10, CIFAR-100, and Flower102. We compare the performance of SGD-GALA, ADAM-GALA against SGD, Adam, AdGD, Mechanic, D-Adaptation(Adam) and Prodigy. We initialize each algorithm with learning rates $[1, 10^{-1}, 10^{-2}, 10^{-3}, 10^{-4}, 10^{-5}, 10^{-8}]$ and execute 3 seeded runs.

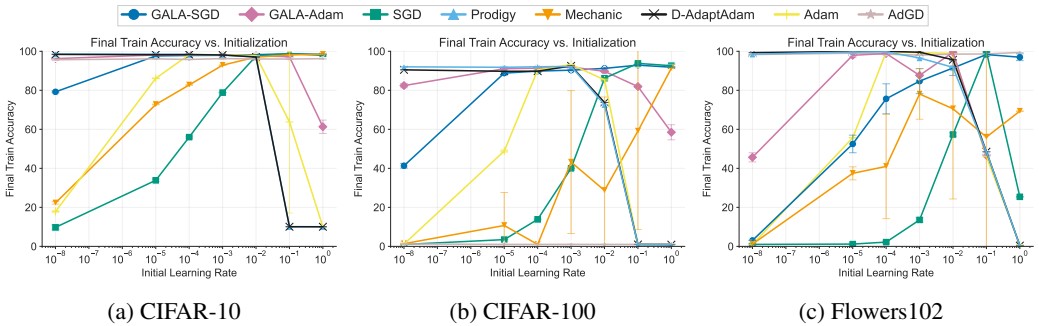

(a) CIFAR-10      (b) CIFAR-100      (c) Flowers102

Figure 7: **Final training accuracy** obtained from different initial learning rates for CIFAR-10, CIFAR-100, and Flower102. We compare the performance of SGD-GALA, ADAM-GALA against SGD, Adam, AdGD, Mechanic, D-Adaptation(Adam) and Prodigy. We initialize each algorithm with learning rates $[1, 10^{-1}, 10^{-2}, 10^{-3}, 10^{-4}, 10^{-5}, 10^{-8}]$ and execute 3 seeded runs.

Across the three different datasets, we observe that our method, SGD-GALA and ADAM-GALA, remains robust with respect to the initial learning rates and has negligible variance due to random seeds. SGD and Adam are sensitive to the choice of the initial learning rate; small learning rate prevents progress while relatively large values yields unpredictable behavior with abrupt changes in performance.

The results on all three datasets (although more pronounced for CIFAR-10 and CIFAR-100) show that Mechanic tends to perform better with larger learning rates, but the variance is high with different random seeds. Interestingly, AdGD fails on CIFAR-100 dataset; as we will discuss later in more detail over the learning rate evolution of the method, this is likely due to the fact that its learning rate becomes too large in some scenarios. For other datasets, AdGD shows consistent convergence across different initialization with great test performance and slightly worse training loss/accuracy. The behavior of parameter-free methods are quite close to each other for all three datasets. Both Prodigy and D-Adaptation show one of the best performances for training and testing when the initial step size is small. however, the performance drops sharply when the initialization is larger than $10^{-2}$.

Compared to Mechanic and AdGD, the variance for different seeds is smaller for SGD-GALA and ADAM-GALA. Among the GALA-variants, SGD-GALA performs better than ADAM-GALA for larger learning rates.

### E.2.2 VIT ON TINY-IMAGENET-200

The robustness provided by GALA framework is better displayed with the ViT experiments; SGD-GALA improves the performance of SGD from all initializations and shows almost the same performance for all initializations. ADAM-GALA shows a similar performance improvement

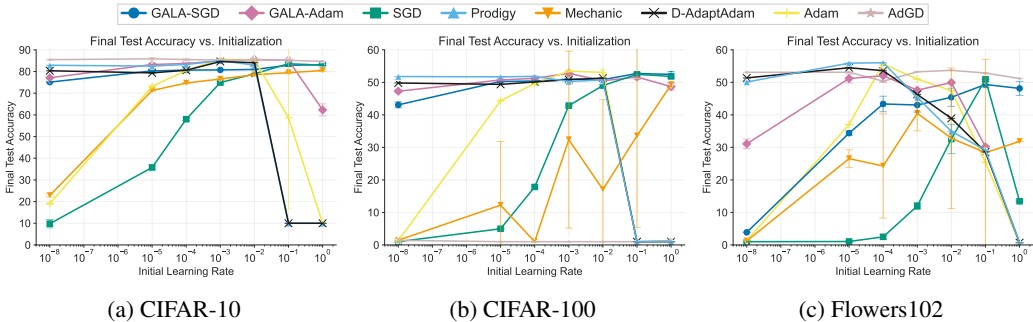

(a) CIFAR-10      (b) CIFAR-100      (c) Flowers102

Figure 8: **Final test accuracy** obtained from different initial learning rates for CIFAR-10, CIFAR-100, and Flower102. We compare the performance of SGD-GALA, ADAM-GALA against SGD, Adam, AdGD, Mechanic, D-Adaptation(Adam) and Prodigy. We initialize each algorithm with learning rates $[1, 10^{-1}, 10^{-2}, 10^{-3}, 10^{-4}, 10^{-5}, 10^{-8}]$ and execute 3 seeded runs.

with respect to AdamW, particularly for the initialization regimes where it performs poorly, which corresponds to small or large initial values of the learning rate. However, we underline that the best-performing AdamW is better than that of ADAM-GALA.

During training, the best-performing configuration for AdamW is better than that of any other method we tested, for which we use prescribed set of parameters and such a behavior is expected. However, SGD-GALA has the best test accuracy, matching and even surpassiong the test perforamnce of AdamW. An interesting obserbation is that SGD-GALA achieves this performance from any initial learning rate value, which validates our claims on stability and robustness to initialization.

In all our experiments, GALA provide a trade-off between training performance and robustness to initialization; for some experiments performance matches the best-performing algorithm while for all scenarios, GALA variants of standalone method show robustness to variation in initial learning rate. Prodigy has strong performance from small initializations across the board but a sharp degradation happens for learning rates larger than $10^{-3}$.

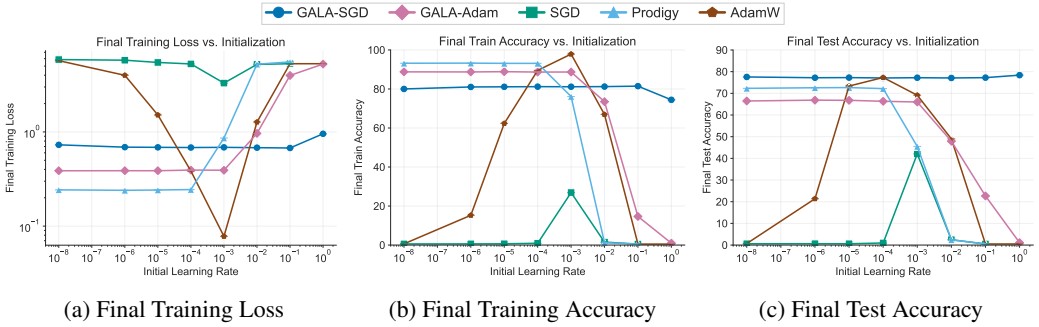

(a) Final Training Loss      (b) Final Training Accuracy      (c) Final Test Accuracy

Figure 9: **ViT (Tiny variant) on Tiny-ImageNet dataset.** We compare SGD-GALA and ADAM-GALA against SGD, AdamW and Prodigy. Each algorithm is initialized with learning rates $[1, 10^{-1}, 10^{-2}, 10^{-3}, 10^{-4}, 10^{-5}, 10^{-6}, 10^{-8}]$ and other parameters are set to default values. The runs are averaged over 2 random seeds.

### E.2.3 LEARNING RATE EVOLUTION

To better understand the convergence behavior of our method, we visualize the learning rate dynamics during training. We demonstrate the evolution of learning rates for SGD-GALA and ADAM-GALA for different models and datasets.

As shown in Figure 10a, the learning rate of SGD-GALA evolves similarly and converges to similar values across a wide range of initialization, excluding extreme cases such as $\eta = 1, 0.1$, or $10^{-8}$. This convergence likely explains the robustness of SGD-GALA to the choice of initial learning rate. Also,

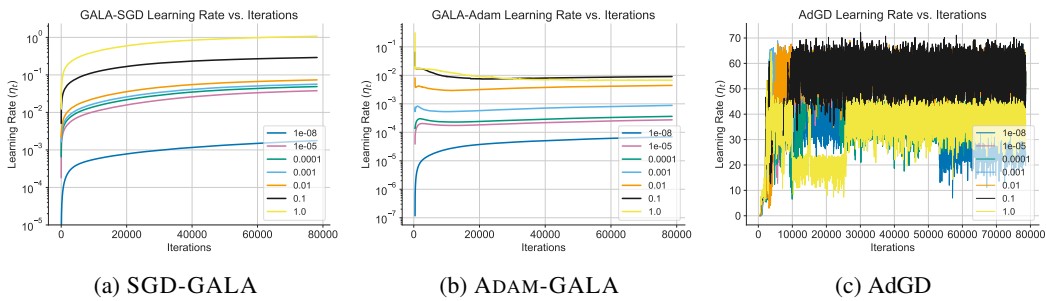

(a) SGD-GALA       (b) ADAM-GALA       (c) AdGD

Figure 10: Comparison of **learning rate evolution** for SGD-GALA, ADAM-GALA and AdGD on the CIFAR-100 dataset, averaged over 3 runs.

in Figure 10b, we observe that, depending on the initial value, the learning rate of ADAM-GALA adapts over time and can both increase and decrease, consistent with the trend seen in Figure 5 on CIFAR-10. In most cases, the learning rate stabilizes between $10^{-2}$ and $10^{-3}$, which roughly corresponds to the best fixed learning rate for Adam according to Figure 8b. By contrast, Figure 10c shows that the learning rate chosen by AdGD tends to oscillate and frequently becomes excessively large, which may contribute to its degraded performance. While AdGD performs competitively on CIFAR-10, its behavior on CIFAR-100 suggests that it may be less robust and that the hyperparameter $\alpha$ in (24) may require retuning for stable performance on new datasets.

Figure 11 shows the learning rate evolution for ViT + Tiny-ImageNet for SGD-GALA and ADAM-GALA. For this setup, the learning rate evolution across different initializations demonstrate a more consistent behavior. Specifically, SGD-GALA converges to the same narrow value range for the learning rate from different initializations between $1$ and $10^{-8}$. While doing so, the learning rate decreases initially and starts increasing for most of the execution. For ADAM-GALA, the behavior seems a bit more complex; the learning rate fluctuates for a while until it attains a monotonic, increasing nature. Note that the learning rate for initializations between $[10^{-2}, 10^{-8}]$ converge to almost the same value with an increasing behavior for the most of the run, while from initial values of $1$ and $0.1$, the learning rate tends to decrease which hurts the performance as shows in the convergence plots.

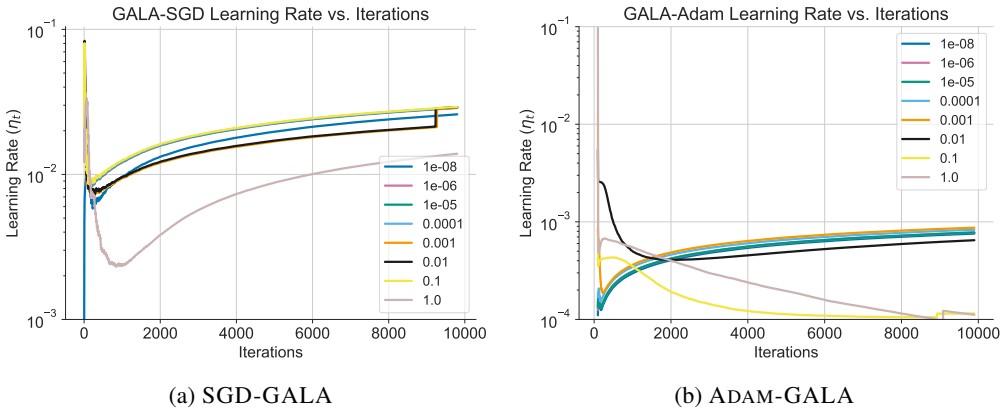

(a) SGD-GALA              (b) ADAM-GALA

Figure 11: Comparison of **learning rate evolution** for SGD-GALA and ADAM-GALA for training ViT-Tiny on Tiny-ImageNet, averaged over 2 runs.

## F    USE OF LARGE LANGUAGE MODELS

We used large language models in limited capacity to help polish phrasing and check grammatical correctness of select parts of the paper.

