# OpenReview forum: "Online Learning-guided Learning Rate Adaptation via Gradient Alignment"
_ICLR.cc/2026/Conference — Submitted to ICLR 2026_

### Official Review · Reviewer_dR1X · 2025-10-23

**Soundness:** 2
**Presentation:** 3
**Contribution:** 1
**Rating:** 2
**Confidence:** 4

**Summary:**

This paper proposes GALA (Gradient Alignment–based Learning Rate Adaptation), an optimizer framework that adaptively adjusts the learning rate by monitoring the alignment between consecutive gradients and estimating local curvature. The authors formulate learning-rate selection as a one-dimensional online learning problem and derive convergence guarantees for a normalized SGD variant. Empirically, they apply GALA to SGD and Adam on several vision benchmarks (CIFAR-10/100, Flower102, Tiny-ImageNet), claiming that GALA improves robustness to initial learning rate choices and achieves competitive accuracy without manual tuning.

**Strengths:**

$\textbf{Reasonable thinking behind Algorithm design}$
* The paper presents a novel conceptual connection between online learning and optimizer dynamics, leveraging regret minimization to adapt step sizes. The proposed framework is theoretically motivated and systematically integrates curvature estimation and gradient alignment.
* The authors provide both theoretical convergence analysis and empirical validation, which makes the contribution reasonably well-rounded.

**Weaknesses:**

$\textbf{Practical algorithm: increased gradient oracle complexity and unfair comparison}$
* Algorithm 1 extends SGD but requires two additional gradient evaluations, an extra full parameter update, and larger memory consumption.
* Consequently, Figure 1 is not a fair comparison, since it plots results against epoch count. A fair evaluation should use computational cost, e.g., total number of gradient queries, as the x-axis.
* The added computational overhead fundamentally weakens the practicality of the proposed method, especially when comparing with single-query optimizers such as SGD or Adam.

---

$\textbf{Algorithm design lacks justification}$
* The introduction of the second local Lipschitz estimate is heuristic: the paper never discusses the estimation error or its impact.
* The design choice $w_{t}=x_{t+1}$  (Sec. 5.1) directly violates the theoretical assumption that $w_{t}$ should lie between $x_{t}$ and $x_{t+1}$.
  * Although this simplification may be intended to save computation or mitigate variance, it breaks the theoretical foundation underlying Lemma 1 and Theorem 1.
  * The authors should clarify whether this approximation still preserves the intended properties or whether it merely serves as a heuristic implementation.

---

$\textbf{Questionable intuition and motivation}$

* The main intuition, monitoring gradient alignment $\Delta_{t}, g_{t}$ as a “useful signal” for learning-rate adjustment, is conceptually debatable.
  * In practice, disagreement between consecutive stochastic gradients is often due to noise, which is known to be beneficial for exploring non-convex landscapes (curvature effects).
  * Completely suppressing this noise, as suggested by the “alignment-only” rule, might reduce exploration and harm convergence in flat or saddle-point regions.

* The choice of Follow-the-Regularized-Leader (FTRL) as the online subroutine appears intuitive rather than principled. The authors do not compare it against other plausible online learners nor discuss its implications.

---

$\textbf{Unfair experimental comparison}$
* Like I mentioned before, the empirical results use epoch count as the horizontal axis, which ignores the doubled number of gradient evaluations in SGD-GALA and ADAM-GALA.
* To be fair, results should be reported with respect to total gradient queries or wall-clock time. Without this normalization, claims of comparable efficiency are not convincing.

---

$\textbf{Limited experimental validation}$
* The empirical section is narrow in scope; only a few architectures (ResNet-18, ViT-Tiny) and small datasets (CIFAR-10/100, Flower102, Tiny-ImageNet) are evaluated.
* The study does not assess GALA’s behavior in larger-scale or more diverse tasks, such as NLP benchmarks or self-supervised pretraining, which would better demonstrate robustness.
* The paper should also include a comparison of computational cost and memory usage between baseline optimizers (SGD, Adam) and their GALA-augmented counterparts (SGD-GALA, ADAM-GALA).

**Questions:**

* In Section 5.1, the practical version of the algorithm appears simplified. However, if I understand correctly, the proposed method still requires two gradient evaluations per parameter update, unlike standard SGD, which needs only one. Could the authors please clarify whether this interpretation is correct?

* Can you provide a comparison of computational cost and memory usage between baseline optimizers (SGD, Adam) and their GALA-augmented counterparts (SGD-GALA, ADAM-GALA)?

---

> ### Author Response · Authors · 2025-11-25
>
> We sincerely thank the reviewer for their detailed feedback. We now address your concerns below.
>
> **W1: Algorithm 1 requires two additional gradient evaluations, an extra full parameter update, and larger memory consumption. In Section 5.1, the practical version of the algorithm appears simplified. However, if I understand correctly, the proposed method still requires two gradient evaluations per parameter update, unlike standard SGD, which needs only one.**
>
> **A1** Thank you for the comment. In our practical implementation of Algorithm 1 (Section 5.1), the method indeed requires one additional gradient evaluation per update compared to standard SGD. While this introduces additional computation and memory usage, we believe this is a reasonable trade-off for improved stability and reduced need for hyperparameter tuning. In contrast, vanilla SGD typically requires careful learning rate tuning to work properly, which can lead to a substantially higher overall cost when accounting for tuning runs. Moreover, the two gradients in our method are evaluated at two different batches, which opens the possibility of leveraging parallel computation to mitigate the overhead. Designing an efficient parallelized implementation is an interesting direction for future work.
>
> ---
> **W2: The introduction of the second local Lipschitz estimate is heuristic: the paper never discusses the estimation error or its impact.**
>
> **A2** Thank you for raising this point. We would like to clarify that the introduction of the second local Lipschitz estimate $\tilde{L}\_t$ is not heuristic but arises directly from the convergence analysis. In particular, Lemma 4 in Appendix C shows that the approximation error of the exponential moving average, $E[\\|m_t  - \nabla F(x_t)\\|]$, is bounded in terms of the local Lipschitz constant $\\tilde{L}\_t$. Hence, this dependence leads to the additional term in the surrogate loss function in (9). In our experiments, we set $w_t=x_{t+1}$, in which case the two Lipschitz estimates $L_t$ and $\tilde{L}_t$ coincide. Therefore, we only maintain a single Lipschitz estimate in practice.
>
> ---
>
> **W3: The design choice $w_t = x_{t+1}$ (Sec. 5.1) directly violates the theoretical assumption that $w_t$ should lie between $x_t$ and $x_{t+1}$. Although this simplification may be intended to save computation or mitigate variance, it breaks the theoretical foundation underlying Lemma 1 and Theorem 1. The authors should clarify whether this approximation still preserves the intended properties or whether it merely serves as a heuristic implementation.**
>
> **A3** Thank you for your question. In our preliminary experiments, we implemented the algorithm using both a randomly interpolated point between $x_t$ and $x_{t+1}$ as well as the simpler choice $w_t =x_{t+1}$, and we observed negligible differences in performance. To reduce memory usage and avoid an additional gradient computation, we therefore adopted the choice $w_t = x_{t+1}$ in our experiments. We have clarified in the revision that it is used as a practical heuristic rather than one required by the theory. Finally, we remark that it is common and often necessary for empirical implementations to introduce practical refinements to theoretical algorithms, even when such modifications lack formal guarantees; the use of learning rate schedulers is such an example. In our view, our adjustments follow the same philosophy and do not undermine the conceptual connection between the theoretical analysis and the empirical method.
>
> ---
> **W4: The main intuition, monitoring gradient alignment as a “useful signal” for learning-rate adjustment, is conceptually debatable. In practice, disagreement between consecutive stochastic gradients is often due to noise, which is known to be beneficial for exploring non-convex landscapes (curvature effects). Completely suppressing this noise, as suggested by the “alignment-only” rule, might reduce exploration and harm convergence in flat or saddle-point regions.**
>
> **A4** We agree that stochastic gradients naturally introduce noise, which can help exploration in nonconvex landscapes. However, this does not contradict our approach. Because the learning rate in our method is updated incrementally via an online learning algorithm, the “alignment-only’’ signal does not fully suppress stochastic variability; rather, it moderates step sizes to prevent instability or divergence. Noise due to mini-batching, sampling (variation amongst component functions) and the data distribution are still at play, which we do not control. We also note that intuition based on gradient disagreement can be difficult to formalize and may be misleading in complex nonconvex settings. For this reason, our paper adopts a more principled perspective: we provide a convergence analysis that captures the behavior of our update rule and supplement it with empirical evidence demonstrating that the method performs well in practice.

---

> > ### Author Response · Authors · 2025-11-25
> >
> > **W5: The choice of Follow-the-Regularized-Leader (FTRL) as the online subroutine appears intuitive rather than principled. The authors do not compare it against other plausible online learners nor discuss its implications.**
> >
> > **A5** Thank you for your question. We respectfully disagree with the reviewer’s characterization. FTRL is a principled and extensively studied online learning algorithm, and it is especially suited for strongly convex loss functions (Orabona, 2019), which is precisely the case in our formulation. Also note that FTRL does not introduce any additional parameters to the step size update process. While alternatives such as Online Gradient Descent (OGD) could also be used, we chose FTRL because it offers favorable empirical performance in our setting and leads to a simple, interpretable update rule for the step size.
> >
> > ---
> > **W6: The empirical section is narrow in scope; The study does not assess GALA’s behavior in larger-scale or more diverse tasks, such as NLP benchmarks or self-supervised pretraining, which would better demonstrate robustness.**
> >
> > **A6** Thank you for the suggestion. We will try to get ImageNet-1K experiments running during the rebuttal period.
> > However, due to tight time constraints and limited computational resources, we may not be able to perform a broader suite of large-scale experiments within the rebuttal window. We view this as an important direction for future work.
> >
> > ---
> > **W7: To be fair, results should be reported with respect to total gradient queries or wall-clock time. The paper should also include a comparison of computational cost and memory usage between baseline optimizers (SGD, Adam) and their GALA-augmented counterparts (SGD-GALA, ADAM-GALA).**
> >
> > **A7** Thank you for this helpful suggestion. Our runtime is longer than methods like Mechanic and Prodigy, but pretty similar to AdGD. Although we make 2 gradient queries per iteration, our runtime is not twice as slow. For instance, the timing on the WandB logs suggests GALA-SGD takes roughly 35-40% more time than SGD for several randomly picked runs and seeds.
> >
> > Regarding memory usage, GALA requires storing one additional stochastic gradient to compute alignment and update the learning rate. This overhead is small and similar in spirit to other adaptive optimizers, including Mechanic, AdGD, and Prodigy (Adam version), many of which maintain additional gradient- or momentum-like buffers.
> >
> > In the meantime, our method is sample-efficient such that we do not “burn” additional samples for our computation, which is often used as a comparison metric in training (very) large-scale models. Let us also note that we are not proposing a purely empirical work where we focus on efficient implementation and code optimization, therefore, it might be unfair to compare custom methods and native Python implementations. We will do a more detailed empirical analysis of runtimes in the final version.

---

> > > ### Comment · Reviewer_dR1X · 2025-11-27
> > >
> > > Thank you for the clarifications. I’ll keep the score for the following reasons.
> > >
> > > ---
> > >
> > > $\textbf{Unfair comparison}$
> > >
> > > The reported performance comparison is unfair and cannot be justified by appealing to “parallel computation.”
> > >
> > > It is unclear why using multiple devices in parallel should be considered comparable to (or more efficient than) using a single device. This does not constitute a fair basis for evaluation.
> > >
> > > ---
> > >
> > > In addition, other concerns have not been adequately addressed, including the unresolved gap between theory and the insufficiently justified algorithmic modifications, as well as the weaknesses in the empirical evaluation.

---

> > > > ### Author Response · Authors · 2025-11-27
> > > >
> > > > "In addition, other concerns have not been adequately addressed, including the unresolved gap between theory and the insufficiently justified algorithmic modifications, as well as the weaknesses in the empirical evaluation."
> > > >
> > > > All of your concerns were addressed in our responses. If you disagree with our explanations, we fully respect that decision. However, stating that “other concerns have not been adequately addressed” suggests that our detailed responses may not have been fully considered.
> > > >
> > > > Finally, we understand if you decide to reject the paper.

---

### Official Review · Reviewer_1jgD · 2025-10-31

**Soundness:** 4
**Presentation:** 3
**Contribution:** 3
**Rating:** 6
**Confidence:** 4

**Summary:**

Authors designed a self-tuning optimization method using the alignment between two consecutive stochastic gradients (à la hypergradient) based on online learning for which they can prove convergence rates for minimizing the gradient norm in expectation for the problem \E_\xi[f(x, \xi)], assuming a stochastic unbiased estimate of the gradient and that every f(\cdot, \xi) is locally smooth among other common assumptions like bounded variance and global gradient Lipschitzness. They use use a surrogate loss inspired by Zhuang et al (2019) for their online learning game on the step size.

**Strengths:**

Authors are able to put together an intricate analysis combining many ideas (online to nonconvex ideas like Cutkosky et al, surrogate losses inspired by Zhuang et al, a hypergradient-like update rule viewed as an online learning algorithm for learning the step size, like in Gao et al and Chu et al do for convex losses...) resulting in provable rates for these kind of consecutive stochastic gradient aligment techniques for learning rate tuning, in the stochastic nonconvex smooth setting with bounded variance.

**Weaknesses:**

The algorithm requires knowledge of the gradient estimates' variance as well as the final iteration T, via $\alpha$.

The extended related work is mostly the same as the one in the main paper. This is too redundant. Also, hypergradient descent literature in the related work section in the appendix was not even mentioned in the main paper. Given how relevant and related this is to this paper, including the theory works of Gao et al (2024) and Chu et al (2025), I'd urge the authors to have this paragraph in the main paper. I trust this is currently in the supplementary material for strategic purposes only (unfortunately those kind of things happen under the incentives of our current review system).

The experiments do not compare with any of the papers in the hypergradient descent literature which directly or indirectly inspired this work. These prior papers (cited in the present paper) already show an adaptivity and improvement over SGD and Adam when computing its heuristic on top of them (e.g. Baydin et al (2018)). I consider the main contribution of this work to be on the theory side but still the experiments should be fair and compare to some of these prior approaches. Figures 1 and 2 are misleading because they are highlighting an adaptivity phenomenon already present in the hypergradient literature. Claiming novelty on this empirical phenomenon is misleading.

**Questions:**

I am not fully sure whether the point of this paper (within the context of what is currently known) is beyond the following or whether there is anything else that I missed, so let me state what I gathered and you can correct me if I am wrong: I suppose that since it was proven by Gao et al. (2024) and Chu et al. (2025) that hypergradient heuristics could be made formal and convergence could be shown by analyzing the learning-rate-tuning hypergradient rule via online learning, for the optimization of convex problems, authors wanted to investigate whether some analogous phenomenon would be possible in stochastic nonconvex problems, by exploiting recent developments that allow to show good rates of convergence via online learning and tricks like the one about sampling a random point w in between x_t and x_{t+1}. Is there any adaptivity that is gained from this algorithm beyond making the hypergradient-like heuristic provably work? Adaptivity to the unknown local gradient Lipschitz constant perhaps?

I'd suggest to write gradient Lipschitz estimate in the many places where it only says Lipschitz estimate referring to the Lipschitzness of the gradient

---

> ### Author Response · Authors · 2025-11-25
>
> We greatly appreciate the reviewer's effort in evaluating our paper and providing detailed feedback. We now address your concerns below.
>
> **W1: The algorithm requires knowledge of the gradient estimates' variance as well as the final iteration T, via $\alpha$.**
>
> **R1** Thank you for raising this point. In Theorem 1, the choice of $\alpha$ is made to balance the terms in (16) and obtain the best dependence on both $T$ and $\sigma$. We note that assuming knowledge of the total number of iterations $T$ is standard in the literature and typically not restrictive in practice. If $\sigma$ is unknown, one could choose  $\alpha = 1/\sqrt{T}$ and still obtain the same convergence rate, albeit with a worse dependence on $\sigma$. Finally, we think determining $\alpha$ adaptively seems to be a more challenging theoretical problem than choosing the learning rate, and addressing this in full generality remains an interesting direction for future work. We added a remark after Theorem 1 to clarify this point.
>
> ---
> **W2: The extended related work is mostly the same as the one in the main paper. Also, hypergradient descent literature was not mentioned in the main paper.**
>
> **R2** Thank you for pointing this out. Due to the page limit in the initial submission, we moved several portions of the related work discussion to the appendix to save space. With the additional page allowed for the rebuttal, we have now integrated this extended discussion back into the main text for clarity and completeness.
>
> ---
> **W3: The experiments do not compare with any of the papers in the hypergradient descent literature which directly or indirectly inspired this work. Figures 1 and 2 are misleading because they are highlighting an adaptivity phenomenon already present in the hypergradient literature.**
>
> **R3** Thank you for this detailed comment. Following your suggestion, we have included comparisons with hypgradient descent (HGD) in the updated manuscript. Unlike our method, HGD requires tuning an additional hyperparameter: hypergradient learning rate, which is used for the gradient update on the _actual learning rate_ $\eta_t$. We trained ResNet-18 models on CIFAR-10 and Flower-102 datasets and selected initial value for the actual learning rate $\eta_0$ and the hypergradient learning rate $\gamma_t$ from the set $\\{1, 10^{-1}, 10^{-2}, 10^{-3}, 10^{-4}, 10^{-5}\\}$ (a total of 36 runs per dataset). On both datasets, HGD diverged (from all initial learning rates) when the hypergradient learning rate is relatively large ($\{1, 0.1\}$ for CIFAR-10 and $1$ for Flower102). The behavior is more stable as the hypergradient learning rate gets smaller; however, this would impact the convergence speed of the algorithm as the learning rate $\eta_t$ evolves slowly, causing training to stagnate for most initializations. The updated manuscript includes plots comparing HGD with our method.
>
> We also emphasize that, although related, our algorithm is conceptually and technically distinct from hypergradient-based approaches. Hyerpgradient descent heuristically updates the learning rate via gradient descent on the learning-rate parameter. In contrast, our method is derived from a principled online-learning formulation: we construct a surrogate loss for the learning-rate updates (following the approach of Zhuang et al., 2019, and the online-to-nonconvex conversion of Cutkosky et al., 2023) and apply an FTRL update. This perspective is what enables our non-asymptotic convergence guarantees, which are not available for prior hypergradient methods.
>
> Finally, our intention in Figures 1 and 2 is not to claim that adaptivity is novel per se, but to illustrate the adaptivity properties of our specific update rule. We will revise the text to avoid any implication that this empirical behavior is unique to our method.

---

> ### Author Response · Authors · 2025-11-25
>
> **Q1: I am not fully sure whether the point of this paper (within the context of what is currently known) is beyond the following or whether there is anything else that I missed, so let me state what I gathered and you can correct me if I am wrong: Since it was proven by Gao et al. (2024) and Chu et al. (2025) that hypergradient heuristics could be made formal and convergence could be shown by analyzing the learning-rate-tuning hypergradient rule via online learning, for the optimization of convex problems, authors wanted to investigate whether some analogous phenomenon would be possible in stochastic nonconvex problems.  Is there any adaptivity that is gained from this algorithm beyond making the hypergradient-like heuristic provably work?**
>
> **A1** Thank you for your detailed questions and for summarizing your understanding of our goals. We clarify the distinctions and motivations below.
>
> As discussed in our prior responses, our algorithm is conceptually and technically different from hypergradient descent. While hypergradient methods adjust the learning rate using a gradient-descent–type update on the step-size parameter, our approach constructs a surrogate loss for the learning rate and applies an FTRL update, which is inspired by Zhuang et al. (2019) and the online-to-nonconvex conversion of Cutkosky et al. (2023). This reformulation is what enables our non-asymptotic convergence guarantees in the stochastic nonconvex setting. That said, we agree that it is also an interesting direction to provide a convergence guarantee for hypergradient descent methods in stochastic nonconvex settings.
>
> Regarding your summary: you are correct that one motivation of our work is to investigate whether the online-learning viewpoint that has proven successful in convex settings (e.g., Gao et al., 2024; Chu et al., 2025) can be extended to the stochastic nonconvex regime. However, this extension is far from straightforward. The techniques in Gao et al. and Chu et al. rely crucially on properties of convexity and deterministic gradients, and it is unclear how their analyses could be adapted to handle stochastic gradients or the lack of convexity. Our convergence results require different analytical tools and a different construction of the loss function for the online-learning problem.
>
> Regarding adaptivity: our algorithm does indeed provide additional adaptivity to the local gradient Lipschitz constant, since the convergence rate in Theorem 1 depends on the local Lipschitz gradient estimates instead of global Lipschitz constants.
>
> **Q2: I'd suggest to write gradient Lipschitz estimate in the many places where it only says Lipschitz estimate referring to the Lipschitzness of the gradient.**
>
> **A2** Thank you for the suggestion. We have updated the manuscript to use the more precise term "gradient Lipschitz estimate" when referring to our estimate for the Lipschitz continuity of the gradient.

---

### Official Review · Reviewer_f2NE · 2025-11-01

**Soundness:** 4
**Presentation:** 4
**Contribution:** 2
**Rating:** 4
**Confidence:** 4

**Summary:**

This paper proposes GALA (Gradient Alignment–based Learning-rate Adaptation): a framework that updates the scalar learning rate via an online learner using a surrogate loss built from (i) alignment between consecutive stochastic gradients evaluated at an interpolated point and the current point, and (ii) a local Lipschitz estimate along the update segment. Theoretical analysis is provided for normalized SGD with momentum using a modified surrogate (Eq. (9)) and yields a rate depending on the average/max local Lipschitz estimates and the regret of the online learner (Theorem 1). Empirically, the authors add a GALA-style update on top of SGD and Adam across CIFAR-10/100, Flowers102, and a ViT-Tiny finetune; figures show improved robustness to the initial learning rate in some regimes.

**Strengths:**

1. The paper presents a principled intuition by casting learning-rate selection as a one-dimensional online learning problem with an explicit surrogate derived from the gradient alignment identity and a local curvature penalty (Eq. (5)), leading to clean, closed-form FTRL updates (Eq. (7)).

2. The theoretical analysis targets normalized SGD with momentum and clearly shows the dependence on an along-trajectory smoothness notion (Theorem 1), complemented by a regret bound (Lemma 2).

3. The algorithmic presentation is clear and provides an accessible description of the optimistic FTRL framework with meaningful intuition.

4. The proposed method is of practical interest, since the idea of using gradient alignment to dynamically increase or decrease the learning rate is intuitive, easy to implement, and addresses a relevant problem of robustness to learning rate selection.

**Weaknesses:**

Regarding the theoretical results:

1. Although not explicitly framed as assumptions, strong assumptions are required for Theorem 1's bound to converge. (1) Theorem 1’s convergence requires the online regret on a surrogate whose quadratic term is strongly convex if $L_t$ and $\tilde L_t$ admit uniform lower bounds, which is restrictive in stochastic nonconvex settings and stronger than assuming only $F$ is $L$-smooth. This basically requires the function $f(\cdot; \xi)$ to be strongly-convex, and cannot be flat locally. (2) For the bound in Theorem 1 to converge, it also relies on sample-wise smoothness of $f(\cdot;\xi)$ via $L_t,\tilde L_t$, which can be substantially stronger than mere smoothness of $F$.

2. The claim that “the convergence rate in Theorem 1 is in terms of the average and maximum Lipschitz estimates, which can be much smaller than the global constant $L$” is not justified. The quantity $L_T^{\text{avg}}$ aggregates per-sample local secant constants and need not be smaller than the global $L$ for $F$; it can even be larger, depending on the noise realizations, unless additional per-sample regularity is assumed. Clearly per-sample smoothness of $f(\cdot, \xi)$ can be larger than the smoothness parameter of the average $F(\cdot)$. Even consider locally, $|| \nabla^2_x f(x, \xi) ||_2$ can be larger than $|| \nabla^2_x F(x) ||_2$. If the argument is not about sample-wise smoothness, but path-wise smoothness is smaller than the global smoothness, then I guess vanilla SGD can also have similar bounds. The intuition is to telescope the local path-wise smoothness inequality instead of the global smoothness. I am not sure if this could work rigorously in math, but intuitively we only care about the smoothness of the function along the optimization path even for SGD, right? What is the fundamental difference here?

3. The algorithm is proposed to require less tuning, “parameter-free” in a sense. However the theoretical rate in Theorem 1 fixes the momentum parameter using $\alpha=\min\{1/(\sigma\sqrt{T}),1\}$, which depends on both $T$ and $\sigma$.

I find the gaps between theory and practice to be significant, leading to the question of whether the theory really explains the robustness of the algorithm in practice:

4. The analyzed algorithm (normalized SGD with momentum) differs from the implemented versions (GALA-SGD and GALA-Adam). The experiments modify Algorithm 1 in key ways: setting $w_t=x_{t+1}$ (no random interpolation) and using the same minibatch to compute both gradients in the alignment term. This contradicts with the major motivation given in Eq. (4). The analysis does not even cover vanilla SGD.

5. Lemma 2’s regret bound relies on $\eta\in[0,\eta_{\max}]$ (bounded domain via clipping). The paper explicitly states that in practice the clipping is removed, i.e., $\eta_{\max}\to\infty$. However, in this case the regret analysis in Lemma 2 becomes vacuous.

Regarding the experiments:

6. GALA uses an extra gradient call per iteration to estimate the Lipschitz constant, making one “iteration” of GALA more expensive than an iteration of vanilla SGD or Adam. Therefore, in Figure 1, comparisons versus iteration count (or epoch count) instead of number of gradient calls would favor GALA, leading to possibly unfair comparison. Also, in Figure 1 (b), there seems to be a bad initial learning rate for GALA-Adam that causes high training loss.

7. In Figure 2, Prodigy and D-AdaptAdam are empirically stable across a wide learning-rate sweep and often match or exceed GALA variants.

Minor issues:

8. In line 206, the definition of $g'_t$ should be $\nabla f(x_t; \xi'_t)$ instead of $\nabla f(x_t; \xi_t)$?

**Questions:**

Please refer to the weaknesses section.

---

> ### Author Response · Authors · 2025-11-25
> **Response to Reviewer f2NE**
>
> **Weaknesses:**
>
> 1. *Although not explicitly framed as assumptions, strong assumptions are required...*:
>
> Thank you for bringing up this point. The first issue can be resolved by replacing $L_t$ with $\max\{L_t, M\}$ in the surrogate loss function in (9) for some small constant $M > 0$. The role of $M$ is analogous to the $\delta$ parameter commonly used in Adam and AdaGrad to ensure well-defined convergence bounds (i.e, safeguarding against very small or zero gradients at early iterations). With this modification, the assumption on $M_{\text{avg}}$ is no longer needed, since the average of $\max\{L_t, M\}$ is automatically lower bounded by $M$, ensuring the surrogate loss functions are strongly convex **without any additional assumptions**. We have updated the manuscript and the corresponding bounds accordingly.
>
> We would also like to clarify the nature of the $M_{\text{avg}}$ assumption. Importantly, we do not assume a lower bound on the true Lipschitz constant of the gradient. Instead, we assume that the average of the empirical local estimates $L_t = \lVert \nabla f(x_{t+1}, \xi_{t+1}) - \nabla f(x_t, \xi_{t+1}) \rVert / \lVert x_{t+1} - x_t \rVert$ is bounded away from zero. The purpose of this mild assumption essentially means we need **at least one** such an estimate to be non-zero to guarantee that the step-size update and the resulting convergence bound are well defined. In addition, since the assumption concerns only the average of the empirical estimates, it does not impose a uniform lower bound on $L_t$, which means it does not rule out the existence of negative eigenvalues or flat regions of the objective. As noted before, the aforementioned fix removes the need for this condition entirely.
>
> We totally agree with your statement that the convergence directly depends on the behavior of $L_t$ sequence, which is exactly what we aimed for: a completely trajectory-dependent convergence bound. As we partially explained above, $L_t$ is the ratio between norm of the observed gradient difference and the distance between query points of these gradients and we neither claim not assume that such a quantity is the local Lipschitz constant of $f(\cdot;\xi)$. It is merely a quantity that estimates how the curvature might behave locally, and does not impose constraints on the actual local curvature (smoothness).
>
> 2. *The claim that “the convergence rate in Theorem 1 is in terms of the average and maximum Lipschitz estimates,...*:
>
> We appreciate your critical evaluation; we should have been clearer with our statements. You are correct that $L_t$ could be larger than $L$ at any point, and we do not have a bound on this variation unless we make stronger assumptions. First of all, all our proofs and derivations hold if we had replaced $L_t$ with $L$, which is the common assumption for standard analysis techniques. To make the algorithm parameter-free and tuning-efficient, our proposal is to use local estimates of $L$, which comes with clear empirical benefits, too.
>
> As you underline, our claim is related to a notion of path-wise smoothness and therefore, our convergence rate is path-dependent. This is similar to data-dependent bounds of AdaGrad, which depend on the sum of norms of the gradient, which could be larger than the Lipschitz smoothness constant (as smoothness bounds the (spectral) norm of the Hessian, and the gradient could be unbounded). We will clarify this phenomenon and explicitly discuss your example in the updated version.
>
> In fact, vanilla SGD cannot have similar bounds (with respect to local Lipschitz estimates) as it is not equipped with the necessary step size modifications to exploit the path-wise behavior. SGD and its variants have a convergence bound that only depends on the global smoothness. Even if one replaces the global smoothness inequality with the local one, the step size and other parameters like momentum do not adopt any adaptive mechanism that could balance (or telescope) the error terms with the local Lipschitz estimates (now the terms become time dependent, so should the step size). We need to modify the SGD step size and replace it with an “appropriate” adaptive one, but unfortunately, this would come with serious measurability issues in the analysis (please refer to [Malitsky and Mishchenko, 2020] for a detailed discussion on the issues for such a modification). Similarly, AdaGrad and Adam would not have $L_t$-dependent bounds for the same reason. Their adaptive step sizes do not accumulate appropriate quantities to reflect the path-wise curvature adaptation.

---

> > ### Author Response · Authors · 2025-11-25
> > **Response to Reviewer f2NE (contd.)**
> >
> > 3. *The algorithm is proposed to require less tuning, “parameter-free” in a sense. However, the theoretical rate in Theorem 1 fixes the momentum parameter using $\alpha=\min{1/(\sigma\sqrt{T}),1}$, which depends on both $T$ and $\sigma$*:
> >
> > Thank you for raising this point. In Theorem 1, the choice of $\alpha$ is made to balance the terms in (16) and obtain the best dependence on both $T$ and $\sigma$. We note that assuming knowledge of the total number of iterations $T$ is standard in the literature and typically not restrictive in practice. If $\sigma$ is unknown, one could choose  $\alpha = 1/\sqrt{T}$ and still obtain the same convergence rate, albeit with a worse dependence on $\sigma$. Finally, we think determining $\alpha$ adaptively seems to be a more challenging theoretical problem than choosing the learning rate, and addressing this in full generality remains an interesting direction for future work. We added a remark after Theorem 1 to clarify this point.
> >
> > 4. *The analyzed algorithm (normalized SGD with momentum) differs from the implemented versions (GALA-SGD and GALA-Adam)...*:
> >
> > As we discuss in the experimental section, we have two main modifications, which are underlined here by our reviewer. We appreciate the reviewer's attention to theoretical details; however, we respectfully disagree with the claim that the theoretical and empirical versions of our method are disconnected. In our implementation, we did test the algorithm exactly as specified in the theory using a randomly interpolated point. We also evaluate other natural alternatives, such as using $x_{t+1}$, using $x_t$, and using the midpoint $(x_t+x_{t+1})/2$. In all cases, the practical differences were negligible; hence, for efficiency, we picked the next iterate as the candidate point.
> >
> > Your comment about bias is valid, and we discuss this choice in the experiments section, where we refer to another algorithm (AdGD) that uses the same modification, for very similar reasons. We implemented both versions to observe the practical differences. The major reason for our practical modification is that reusing the batch for the inner product term preserves the “signal”; the inner product between **gradients from different batches** is affected by the variance due to batching, and it would be difficult to rely on this estimate when noise dominates. On the other hand, the inner product w.r.t. **the same batch** measures alignment of the gradients on the same component function, and it is a more reliable measure of signal. It is fairly common to propose modifications to base algorithms for improved performance in practice, and our approach is no different in our opinion.
> >
> > For instance, AdGD would perform noticeably poorly without an additional parameter that the authors heuristically proposed for empirical performance (we use it as they describe in their paper). This does not diminish the value and contributions of that work, and in fact, shows the rigor the authors of that paper put into the empirical study of their methodology. Similarly, our approach is theoretically principled and relies on the analysis of a generalized version of SGD that forms the backbone of the framework. We modified it following empirical observations, and we provide rigorous theoretical reasoning for such changes.
> >
> > Our analysis would extend to vanilla SGD, but the norm of the gradients would appear in the convergence bound. Normalization is a way of removing gradient norms from the bounds.
> >
> > 5. *Lemma 2’s regret bound relies on $\eta\in[0,\eta_{\max}]$ (bounded domain via clipping)...*:
> >
> > As we point out in the remarks in the manuscript, $\eta_{\max}$ is not a parameter in a true sense. As our reviewer would agree, online algorithms typically operate on bounded domains. There are papers that propose online learning algorithms for unbounded domains, but those algorithms are quite complicated and have their own set of parameters. Therefore, we use $\eta_{\max}$ for well-defined and clean online problem definition, and we never project the step size update onto the range $[0,\eta_{\max}]$. In that case, we can treat $\eta_{\max} = \max_t \eta_t$, which in practice remains bounded as we display in Figures 4 and 9.

---

> > > ### Author Response · Authors · 2025-11-25
> > > **Response to Reviewer f2NE (contd.)**
> > >
> > > 6. *GALA uses an extra gradient call per iteration to estimate the Lipschitz constant, making one “iteration” of GALA more expensive ...*:
> > >
> > > Our runtime is longer than methods like Mechanic and Prodigy, but pretty similar to AdGD. Although we make 2 gradient calls per iteration, our runtime is not twice as slow. For instance, the timing on the WandB logs suggests GALA-SGD takes roughly 35-40% more time than SGD for a couple of randomly picked runs and seeds. In the meantime, our method is sample-efficient such that we do not “burn” additional samples for our computation, which is often used as a comparison metric in training large-scale models.
> > >
> > > Moreover, the two gradients in our method are evaluated at two different batches, which opens the possibility of leveraging parallel computation to mitigate the overhead. Designing an efficient parallelized implementation is an interesting direction for future work. Let us also note that we are not proposing a purely empirical work where we focus on efficient implementation and code optimization. Therefore, it might be unfair to compare custom methods and native Python implementations. We will do a more detailed empirical analysis of runtimes.
> > >
> > > Our framework improves robustness with respect to standalone algorithms, but it cannot completely guarantee a perfect solution (similar to comparable methods like Mechanic, AdGD, Prodigy, D-Adaptation). A way to theoretically explain this is that the initial step size “appears” in the convergence rate as $L_0$ in the regret. Therefore, a very large value of $\eta_0$ could be mitigated by $L_0$ as the iterate difference in the denominator scales down a large change in the initial gradients. This still cannot rule out relatively poor performance, but it is clearly a significant improvement over SGD and Adam bounds. (Please find a relevant discussion in Remarks 5 and 6).
> > >
> > > A very important perspective to this discussion is how we should measure the total cost of training. Although there is no objective measure for evaluating the time spent on the parameter sweep, we believe we should consider this while comparing methods and their potential value. Our method fosters comparably good performance from a wide range of initial values, and hence, the GALA version of the algorithms would need fewer rounds of parameter sweep, cutting the cost of tuning. In simple words, our method spends more time per iteration but requires far fewer rounds of tuning.
> > >
> > >
> > > 7. *In Figure 2, Prodigy and D-AdaptAdam are empirically stable across a wide learning-rate sweep and often match or exceed GALA variants.*:
> > >
> > > We particularly and properly acknowledge their good performance in the paper, and they have their own novelties when it comes to robustness. However, they are very unstable for large initial values, where the performance sharply drops, but our algorithm is visibly more robust in that regime. We do not claim that our algorithm outperforms all existing ones in every aspect, which would be unreasonable. We tried to design several setups with different models and datasets to paint a (more) complete picture so that we can identify regimes and perspectives where our method has clear advantages, while other methods have their own strengths in other domains.
> > >
> > >
> > > 8. *In line 206, the definition of $g'_t$ should be $\nabla f(x_t; \xi'_t)$ instead of $\nabla f(x_t; \xi_t)$?*:
> > >
> > > The reviewer is right, and we have corrected this mistake in the manuscript. Thank you for pointing this out to us.

---

### Official Review · Reviewer_KBHb · 2025-11-03

**Soundness:** 2
**Presentation:** 3
**Contribution:** 2
**Rating:** 4
**Confidence:** 3

**Summary:**

The paper presents a principled online-learning-based approach to estimation of the best stepsize to loss minimization. The authors define a sequence of surrogate loss functions and apply FTRL to the obtained regret minimization problem. The authors study the obtained method when applied to the iterates of normalized-SGD, and under a few assumptions, including lower-bounded Lipschitz constants (which I argue is problematic), they show convergence guarantees matching those of methods that require learning rate tuning. They also test their method on vision problems.

**Strengths:**

1. The paper discusses the related work in details and attributes the ideas properly.

2. The setting for the experiments is reasonable, though limited to the vision datasets. At least I appreciate that the authors didn't stop at running CIFAR-10 evaluations.

3. The authors compared to quite a few other adaptive optimizers, which helps show the strength of the evaluations.

4. The practical implementation does not use many extra hyperparameters, and the method doesn't seem to be sensitive to the extra ones, namely $\delta$ and $\eta_0$.

**Weaknesses:**

1. The assumption that the averages of Lipschitz constants are lower bounded is essentially requiring the eigenvalues of the Hessian matrix to be separated from 0, which means no saddle points. I think it is very strong and should be avoided if the authors are truly interested in the nonconvex optimization. As a result, I don't think combining Theorem 1 and Lemma 2 is a valid way to compare to the results of Cutkosky and Mehta (2020). Similarly, the resuts can't be compared to those of AdaGrad.

2. The actual implementation of the method is quite different from the theory. Most importantly, the authors abandon one of the key ideas of estimating the gradient product using a randomly sampled interpolated point. Moreover, the authors reuse the minibatches, which means that the implemented method doesn't use unbiased estimates. I find these two changes quite important to claim that there is little relation between the method studied theoretically and the one studied empirically.

3. In addition, the proposed method requires two gradients to be computed, which makes it significantly more expensive than other approaches such as Mechanic and Prodigy.

4. The numerical results are not presented in a clear way. I couldn't read Figure 1. While the GALA versions of the methods appear to be better, it's very messy, and particularly unclear at the end of the curves. The latter issue could be fixed by using log scale on the y-axis, but it would also be nice if there was a way to understand which line uses which learning rate. On top of that, I can see that in Figure 1 (a) the best method appears to be SGD rather than GALA-SGD. Finally, the quantities plotted in Figures 2 and 3 aren't very important to be placed in the main body in my opinion. While I understand the need to test sensitivity to the initial stepsize estimate, I think it's the kind of evaluation that be better placed in the appendix.

5. While I understand the authors may have a limited budget for GPUs, I think it would be reasonable to try the tuned methods on full ImageNet-1k instead of Tiny-ImageNet.

Minor comment: please proofread the citations, "Luís B Almeida" should be "Luís B. Almeida", "José D Amaral" should be "José D. Amaral", etc.

**Questions:**

1. How does your method compare to others in terms of run-time?
2. Why are you applying FTRL and not other methods? I understand that you wanted theoretical guarantees, but wouldn't an Adam version of the method be more practical?
3. Some implementation details seem to be missing, for instance, you didn't specify which augmentations and patch sizes you used in ViT (i.e., was it ViT_tiny_patch16_224 and did you use aa="rand-m15-n2" or something else?). Did you clip the gradient norms as usually done when training ViT?

---

> ### Author Response · Authors · 2025-11-25
> **Response to Reviewer KBHb**
>
> We thank our reviewer for their comments, feedbacks and suggestions. Please find our responses below:
>
> **Weaknesses:**
> 1. *The assumption that the averages of Lipschitz constants are lower bounded ...*:
>
> Thank you for bringing up this point. This issue can be resolved by replacing $L_t$ with $\max\{L_t, M\}$ in the surrogate loss function in (9) for some small constant $M > 0$. The role of $M$ is analogous to the $\delta$ parameter commonly used in Adam and AdaGrad to ensure well-defined convergence bounds (i.e, safeguarding against very small or zero gradients at early iterations). With this modification, the assumption on $M_{\text{avg}}$ is no longer needed, since the average of $\max\{L_t, M\}$ is automatically lower bounded by $M$, **without any additional assumptions**. This matches the setting studied in the prior work of Cutkosky and Mehta (2020) as well as the standard AdaGrad analysis. We have updated the manuscript and the corresponding bounds accordingly.
>
> 2. *The actual implementation of the method is quite different from the theory...*:
>
> We appreciate the reviewer's attention to the theoretical details; however, we respectfully disagree with the claim that the theoretical and empirical versions of our method are disconnected. In our implementation, we did test the algorithm exactly as specified in the theory using a randomly interpolated point. We also evaluate other natural alternatives, such as using $x_{t+1}$, using $x_t$, and using the midpoint $(x_t+x_{t+1})/2$. In all cases, the practical differences were negligible; hence, for efficiency, we picked the next iterate as the candidate point.
>
> Your comment about bias is valid, and we discuss this design choice in the experiments section. The same modification is used in AdGD for similar reasons. We implemented both versions to observe the practical differences. The major reason for our practical modification is that reusing the batch for the inner product term preserves the “signal”; the inner product between **gradients from different batches** is affected by the variance due to sampling, and it could be difficult to rely on this estimate when noise dominates. On the other hand, the inner product w.r.t **the same batch** measures alignment of the gradients on the same component function, and it is a more reliable measure of signal.
>
> Finally, we remark that it is common and often necessary for empirical implementations to introduce practical refinements to theoretical algorithms, even when such modifications lack formal guarantees; the use of learning rate schedulers is a standard example. In our view, our adjustments follow the same philosophy and do not undermine the conceptual connection between the theoretical analysis and the empirical method.
>
>
> We would also like to clarify the nature of the $M_{\text{avg}}$ assumption. Importantly, we do not assume a lower bound on the true Lipschitz constant of the gradient. Instead, we assume that the average of the empirical local estimates $L_t = \lVert \nabla f(x_{t+1}, \xi_{t+1}) - \nabla f(x_t, \xi_{t+1}) \rVert / \lVert x_{t+1} - x_t \rVert$ is bounded away from zero. The purpose of this mild assumption essentially means we need **at least one** such an estimate to be non-zero to guarantee that the step-size update and the resulting convergence bound are well defined. The aforementioned fix removes the need for this condition entirely.
>
> Regarding the concern about Hessian eigenvalues and saddle points, we think there may be a misunderstanding. A lower bound on the local Lipschitz constant (largest eigenvalue) does not exclude the existence of negative eigenvalues in the Hessian, and thus our assumption does not rule out saddle points in the problem.
>
> 3. *In addition, the proposed method requires two gradients to be computed, which makes it significantly more expensive...*:
>
> Both methods are important contributions to the literature with their own principled approaches and novelties, and we identified scenarios where our method has practical advantages over theirs in terms of robustness to initializations. For instance, for relatively large initializations, our method performs better than Prodigy, while outperforming Mechanic for smaller learning rates. Although there is no objective measure for including the cost of tuning into the total cost of the optimization process and the parameter sweep, we believe we should also consider this when comparing methods and their potential value. Let us also underline that the convergence guarantees for Mechanic and Prodigy are only applicable to **convex** functions, whereas our method has guarantees for non-convex functions, which we believe is important in terms of novelty and technical rigor.

---

> > ### Author Response · Authors · 2025-11-25
> > **Response to Reviewer KBHb (contd.)**
> >
> > 4. *The numerical results are not presented in a clear way...*:
> >
> > We believe the reviewer missed the main point of the paper, which we would like to highlight here. We are proposing a paradigm that would ensure reasonable and consistent convergence behavior from a wide range of initialization values for the step size, thereby cutting the cost of tuning. As illustrated in Figure 1, if the initial step size is “bad” (too small or too large), SGD and ADAM can converge very slowly, whereas GALA offers a clear improvement on that front. This point is also explicitly highlighted in Figures 2 and 3, where we show the sensitivity of different optimizers to the initial value of the step size, which could be significant depending on the model and dataset. In Figures 2 and 3, our method is the most robust across all the datasets and models we considered, where we do not see abrupt performance declines. We agree that curves overlap with each other in Figure 1, and hence we will update them with a semi-log scale for a better display of the difference for Figure 1.
> >
> > Let us also underline that there does not exist a reliable way of identifying the interval of step sizes that should yield good performance, a priori, for a given algorithm, model, and dataset. In fact, an initial step size of $1e-5$ might be very small for some scenarios, but it could also yield the best performance for another model and dataset pair. Therefore, our framework is a principled and empirically useful proposal for solving this issue by helping the base algorithm (SGD or Adam in this case) achieve good convergence performance (comparable to best-performing curve) from a wider range of initializations, which would otherwise yield poor performance for the base algorithm.
> >
> > 5. *While I understand the authors may have a limited budget for GPUs, I think it would be reasonable to try the tuned methods on full ImageNet-1k instead of Tiny-ImageNet*:
> >
> > We thank the reviewer for their suggestion. Indeed, we have limited GPU access during deadline periods, so we did our best to diversify the experiments. We will try to get IN-1K experiments running during the rebuttal period.

---

> > > ### Author Response · Authors · 2025-11-25
> > > **Response to Reviewer KBHb (contd.)**
> > >
> > > **Questions**:
> > >
> > > 1. *How does your method compare to others in terms of run-time?*
> > >
> > > Our runtime is longer than methods like Mechanic and Prodigy, but pretty similar to AdGD. Although we make 2 gradient calls per iteration, our runtime is not twice as slow. For instance, the timing on the WandB logs suggests GALA-SGD takes roughly 35-40% more time than SGD for several randomly picked runs and seeds. In the meantime, our method is sample-efficient such that we do not “burn” additional samples for our computation, which is often used as a comparison metric in training (very) large-scale models. Let us also note that we are not proposing a purely empirical work where we focus on efficient implementation and code optimization. Therefore, it might be unfair to compare custom methods and native Python implementations. We will do a more detailed empirical analysis of runtimes in the final version.
> > >
> > > 2. *Why are you applying FTRL and not other methods? I understand that you wanted theoretical guarantees, but wouldn't an Adam version of the method be more practical?*:
> > >
> > > The main reason is that we need an algorithm that achieves the $\log(T)$ regret for a strongly-convex sequence of losses. Adam and AdaGrad do not have explicit $\log(T)$ regret guarantees and, in fact,  Adam is shown to be divergent even for convex losses. Second, we indeed implemented Adam and AdaGrad as online algorithms to understand whether an adaptive online learner helps in any way. For most of our experiments, their behavior was not as stable as FTRL, which was rather surprising to us. As an alternative, one could consider online gradient descent (OGD), but we selected FTRL because of its favorable performance in practice, and the resulting update rule for the step size is quite simple and intuitive.
> > >
> > > 3. *Some implementation details seem to be missing, for instance, you didn't specify which augmentations and patch sizes you used in ViT (i.e., was it ViT_tiny_patch16_224 and did you use aa="rand-m15-n2" or something else?). Did you clip the gradient norms as usually done when training ViT?*:
> > >
> > > Indeed, we used the vit_tiny_patch16_224 from timm. We believe you are referring to augmentations with “aa”. We used the ImageNet auto augment policy of torchvision, i.e., torchvision.transforms.AutoAugmentPolicy.IMAGENET which is different than the “rand-m15-n2” option. “rand-m15-n2” should be equivalent to RandAugment in torchvision terms. We didn’t clip the gradients as we wanted to observe the prescribed behavior for our method, and did the same for all the experiments with ViTs for consistency.

---

> > > > ### Comment · Reviewer_KBHb · 2025-11-26
> > > >
> > > > I thank the authors for their responses.
> > > >
> > > > > This issue can be resolved by replacing $L_t$ with $\max\{L_t, M\}$
> > > >
> > > > I'd like to understand what kind of extra hyperparameter is $M$. Based on the updated theory, it appears that the optimal value of $M$ is an upper bound on the Lipschitz constant $L$, is it right? I don't see any assumptions on the value of $M$, are we free to choose any value?
> > > >
> > > > > we respectfully disagree with the claim that the theoretical and empirical versions of our method are disconnected
> > > >
> > > > As I wrote in my initial review "Most importantly, the authors abandon one of the key ideas of estimating the gradient product using a randomly sampled interpolated point." As far as I can see, this point is valid.
> > > > The problem here that the theoretical version is not what anyone would use due to its extra costs. So even if the theoretical method does work sufficiently well, it'd always be the more practical variants that are used in practice.
> > > >
> > > > To be clear, I don't think it is wrong to search for a practical method by developing a theoretical method and then making it more practical. However, it does make the theory itself less interesting and puts more emphasis on the numerical evaluations, which then need to show the designed practical method is indeed useful in practice. This is not the case in submission, unfortunately, as the experiments are still quite limited.
> > > >
> > > > > the use of learning rate schedulers is a standard example
> > > >
> > > > Actually, there has been research on the close alignment between the theory and the practical behaviour of schedulers, see the work of Schaipp et al. "The Surprising Agreement Between Convex Optimization Theory and Learning-Rate Scheduling for Large Model Training"
> > > >
> > > > > Regarding the concern about Hessian eigenvalues and saddle points, we think there may be a misunderstanding. A lower bound on the local Lipschitz constant (largest eigenvalue) does not exclude the existence of negative eigenvalues in the Hessian, and thus our assumption does not rule out saddle points in the problem.
> > > >
> > > > A lower bound on the local Lipschitz constant implies that $|\lambda_i(\nabla^2 f(x))| \ge \mathrm{const}$ for all eigenvalues $\lambda_i$, incluidng the negative ones. If the Hessian is continuous, this excludes the possibility of negative eigenvalues. If the Hessian is discontinuous, it permits strict saddle points only, which are a smaller issue in nonconvex optimization.
> > > >
> > > > > Let us also underline that the convergence guarantees for Mechanic and Prodigy are only applicable to convex functions, whereas our method has guarantees for non-convex functions, which we believe is important in terms of novelty and technical rigor.
> > > >
> > > > I do appreciate technical rigor and theoretical guarantees, but as mentioned earlier, I see a disconnect between the theory and practice in the submission too.
> > > >
> > > > > We believe the reviewer missed the main point of the paper, which we would like to highlight here
> > > >
> > > > Thank you for clarifying the results presented in Figure 1. I'd like to note that characterizing my review as 'missing the main point' may not be the most productive framing. I did have issues reading the figure, but the main idea of reducing the cost of tuning is quite clear to me.
> > > >
> > > >
> > > > ### Minor
> > > > Please make sure to double check that you're using the validation rather than the test part of the imagenet dataset and that the figure labels capture that. Figure 3 (c) is currently reporting "Final Test Accuracy", which might not be what you're actually measuring if you used the validation part of the dataset.
> > > > In line 838, a small typo "Moreover, Since"
> > > > If I'm not mistaken, the first step in equation (18) is identity rather than inequality

---

### Meta-Review · Area_Chair_qZtV · 2026-01-07

**Summary:**

The reviewers broadly agree that the paper proposes a principled and technically interesting framework (GALA), which formulates learning-rate adaptation as a one-dimensional online learning problem and provides non-asymptotic convergence guarantees in the stochastic nonconvex setting. However, the final recommendation is primarily shaped by a major concern regarding **the theory–practice gap**, namely that the actual implementation of the method differs substantially from the theoretical algorithm. Although the authors provided a detailed explanation of why random interpolation is avoided and why the same minibatch is used to compute both gradients in the alignment term. Nevertheless, no additional wall-clock-time experiments were provided. Moreover, the experiments remain relatively limited, as noted by Reviewers KBHb, f2NE, and dR1X in both the reviews and the rebuttal.

**Reviewer Concerns:**

**Reviewer KBHb’s concerns:**

(1) The concerns regarding lower bounds on the Lipschitz constants were clarified and partially alleviated by introducing a stabilizing (user-defined) constant $M$. However, the follow-up question raised on 26 Nov 2025 concerning how $M$ should be chosen was not addressed.

(2) The concern regarding the theory–practice gap was partially addressed. The authors provided a detailed explanation of why random interpolation is avoided and why the same minibatch is used to compute both gradients in the alignment term. Nevertheless, no additional wall-clock-time experiments were provided, and the experiments are still quite limited, as said by Reviewer KBHb in the rebuttal.

(3) The concerns regarding missing implementation details (ViT configuration, augmentations, clipping) were clearly addressed.

**Reviewer f2NE’s concerns:**

(1) The concerns regarding strong convexity and Lipschitz related assumptions were clarified and partially alleviated by introducing a stabilizing (user-defined) constant $M$.

(2) Claims concerning local/path-dependent smoothness were refined, with the clarification that local estimates/bounds could be larger than the Lipschitz smoothness constant.

(3) The concerns regarding the required knowledge of the gradient estimates' variance as well as the final iteration $T$ were largely addressed. However, the proposed approach still relies on prior knowledge of the total number of iterations $T$.

(4) The concern regarding the theory–practice gap was partially addressed. The authors provided a detailed explanation of why random interpolation is not used and why the same minibatch is employed to compute both gradients in the alignment term. However, no additional wall-clock-time experiments were provided. Moreover, the response to Weakness 5 concerning Lemma 2 does not fully address Reviewer f2NE’s concern about the bounded domain via clipping, as the remark following Lemma 2 discusses the parameter $\eta_0$ rather than $\eta^{\mathrm{max}}$.

**Reviewer 1jgD’s concerns:**

(1) The concerns regarding the hypergradient descent literature were addressed through additional experimental comparisons presented in Fig. 4. The authors also moved the relevant hypergradient descent related works into Section 1.1 (Related Work).

(2) The concerns regarding the knowledge of the gradient estimates' variance as well as the final iteration $T$ were adequately addressed. However, the approach still relies on prior knowledge of the total number of iterations $T$.

**Reviewer dR1X’s concerns:**

(1) The main concern regarding unfair experimental comparisons (epochs vs. gradient queries or wall-clock time) remain unresolved, I agree with Reviewer dR1X that this issue cannot be justified by appealing to “parallel computation.”

(2) The concern regarding the theory–practice gap was partially addressed. I agree with the authors that it is common and often necessary for empirical implementations to introduce practical refinements to theoretical algorithms. However, a more explicit discussion of this gap would be beneficial, e.g., clarifying why Line 6 of Algorithm 1 plays a crucial role in the theoretical analysis.

(3) The concerns regarding larger-scale or more diverse tasks were not addressed. The authors said that they may not be able to perform a broader suite of large-scale experiments within the rebuttal window, due to tight time constraints and limited computational resources.

(4) Other concerns, including the choice of Follow-the-Regularized-Leader (FTRL) and the second local Lipschitz estimate, were addressed.

**Reviewer Scores:**

**Reviewer KBHb**, **Reviewer f2NE**, and **Reviewer dR1X** would likely maintain their scores, as the primary concerns regarding the theory–practice gap were only partially addressed.

**Reviewer 1jgD** would likely maintain a score of 6, as the concerns regarding the hypergradient descent literature were adequately addressed.

---

### Decision · Program_Chairs · 2026-01-26

Reject